# Dopamine System, NMDA Receptor and EGF Family Expressions in Brain Structures of Bl6 and 129Sv Strains Displaying Different Behavioral Adaptation

**DOI:** 10.3390/brainsci11060725

**Published:** 2021-05-29

**Authors:** Jane Varul, Kattri-Liis Eskla, Maria Piirsalu, Jürgen Innos, Mari-Anne Philips, Tanel Visnapuu, Mario Plaas, Eero Vasar

**Affiliations:** 1Department of Physiology, Institute of Biomedicine and Translational Medicine, University of Tartu, 19 Ravila Street, 50411 Tartu, Estonia; kattri-liis.eskla@ut.ee (K.-L.E.); maria.piirsalu@ut.ee (M.P.); innos@ut.ee (J.I.); mari-anne.philips@ut.ee (M.-A.P.); tanel.visnapuu@ut.ee (T.V.); mario.plaas@ut.ee (M.P.); eero.vasar@ut.ee (E.V.); 2Center of Excellence for Genomics and Translational Medicine, Institute of Biomedicine and Translational Medicine, University of Tartu, 19 Ravila Street, 50411 Tartu, Estonia; 3Laboratory Animal Center, Institute of Biomedicine and Translational Medicine, University of Tartu, 14B Ravila Street, 50411 Tartu, Estonia

**Keywords:** 129Sv strain, Bl6 strain, gene expression, adaptation, stress model, dopamine system, EGF family, NMDA receptor

## Abstract

C57BL/6NTac (Bl6) and 129S6/SvEvTac (129Sv) mice display different coping strategies in stressful conditions. Our aim was to evaluate biomarkers related to different adaptation strategies in the brain of male 129Sv and Bl6 mice. We focused on signaling pathways related to the dopamine (DA) system, N-methyl-D-aspartate (NMDA) receptor and epidermal growth factor (EGF) family, shown as the key players in behavioral adaptation. Mice from Bl6 and 129Sv lines were divided into either home cage controls (HCC group) or exposed to repeated motility testing and treated with saline for 11 days (RMT group). Distinct stress responses were reflected in severe body weight loss in 129Sv and the increased exploratory behavior in Bl6 mice. Besides that, amphetamine caused significantly stronger motor stimulation in Bl6. Together with the results from gene expression (particularly *Maob*), this study supports higher baseline activity of DA system in Bl6. Interestingly, the adaptation is reflected with opposite changes of DA markers in dorsal and ventral striatum. In forebrain, stress increased the gene expressions of *Egf-Erbb1* and *Nrg1/Nrg2-Erbb4* pathways more clearly in 129Sv, whereas the corresponding proteins were significantly elevated in Bl6. We suggest that not only inhibited activity of the DA system, but also reduced activity of EGF family and NMDA receptor signaling underlies higher susceptibility to stress in 129Sv. Altogether, this study underlines the better suitability of 129Sv for modelling neuropsychiatric disorders than Bl6.

## 1. Introduction

Mice are widely applied to address the neurobiology of psychiatric disorders in animal models. Both inbred lines, Bl6 and 129Sv, are acknowledged mouse strains in biomedical and transgenic research. It is commonly accepted that Bl6 mice show greater locomotor activity and increased exploratory behavior while 129Sv mice are less active and are more vulnerable to stress [1,2,3,4].

It has been shown that the phenotype differences between the 129Sv and Bl6 strains remain stable in most behavioral tests despite environmental modifications and are actually reinforced by environmental enrichment which induces active stress coping (a coping style that is characterized by trying to escape from stressful situations) in Bl6 and a passive response (helplessness to deal with the stressors) in 129Sv [1,3]. Besides that, Bl6 and 129Sv display dissimilarities in many other physiological and behavioral outcomes. For example, Bl6 mice are more aggressive and demonstrate significant preference for alcohol [3,5]. In Bl6 mice, acute treatment with amphetamine (AMPH), an indirect dopamine (DA) agonist, significantly augments locomotor activity and striatal DA efflux compared to 129Sv, but there are no differences in basal DA levels [6]. The effect of repeated treatment with AMPH on the locomotor activity of Bl6 and 129Sv is more complex. Recently we established two differently responding sub-groups among 129Sv: one responded similarly to the acute AMPH group (weak responders), and in the other one the response was 5-fold augmented (strong responders). Such a vast behavioral variation in the response to AMPH in 129Sv was not evident in Bl6 mice [7]. However, according to literature there might be variations in the genetic and behavioral phenotype among Bl6 and 129Sv sub-strains [8].

Besides that, it is worth mentioning that all 129Sv sub-strains have a frameshift mutation in disrupted-in-schizophrenia-1 (*Disc1*) gene, causing alterations in DA homeostasis [9,10,11]. It is apparent that *Disc1* frameshift mutation alters the function of the DA system in 129Sv mice [12,13]. On the other hand, DA systems play a key role in behavioral adaptation, especially via the nigrostriatal and mesocorticolimbic DA systems [14]. DA systems in the dorsal and ventral striatum are under the influence of cortical glutamatergic projections [15]. Via N-methyl-D-aspartate (NMDA) receptors the glutamatergic system modulates the activity of DA projections originating from the midbrain [16]. It is well established that the interactions between NMDA and DA receptors play a role in behavioral adaptation [17,18]. Abnormal function of these systems may cause serious problems in adaptation to novel challenging environments. For example, inactivation of NMDA receptors on DA neurons led to the development of depression-like symptoms in mice: increased immobility in the forced swim test, decreased social interactions and also reduced saccharin intake [19]. Furthermore, NMDA receptor antagonists are known to induce hypermotility and reverse the motor deficits induced by the pharmacological blockade of DA receptors in rodents [20].

Recently it became clear that other molecular regulators may modulate the interaction between DA and NMDA systems. Among those are epidermal growth factor (EGF) family proteins. Neonatal treatment with EGF and neuregulin 1 (NRG1) produced long-lasting behavioral impairments implicated in schizophrenia-like symptoms [21,22]. NRG1-treated mice exhibited behavioral impairments in prepulse inhibition, latent inhibition and social behaviors alongside with hypersensitivity to methamphetamine [21]. Neonatally EGF-treated animals exhibited persistent hyperdopaminergic abnormalities in the nigrostriatal system while NRG-1 treatment resulted in DA-ergic deficits in the mesocorticolimbic DA system [22]. EGF receptor ERBB1 inhibitors appear to have anti-dopaminergic actions alleviating the deficits in startle response and prepulse inhibition induced by neonatal EGF treatment [23]. NRG1/erb-b2 receptor tyrosine kinase 4 (ERBB4) signaling in DA-ergic axonal projections modulates DA homeostasis, whereas NRG1/ERBB4 signaling in both GABA-ergic interneurons and DA neurons in the midbrain contribute to the modulation of animal behaviors relevant to psychiatric disorders [24,25]. Simultaneously, it appears that EGF family proteins affect glutamatergic neurotransmission as well. Several studies demonstrate that EGF family proteins influence the balance of ligand-receptor signaling required for maintaining optimal glutamatergic synaptic plasticity [26,27,28]. EGF receptor signaling upregulates NMDA receptors through modification of the glutamate ionotropic receptor NMDA type subunit 2B (GluN2B, GRIN2B), and is required for high-frequency stimulation-induced long-term potentiation in the hippocampus [29]. NRG1/ERBB4 signaling regulates glutamatergic plasticity by rapidly increasing extracellular hippocampal DA levels and activation of D4 DA receptors [30].

To explore different adaptive behaviors in these strains, we used two different interventions in mice. One batch of mice (Bl6 and 129Sv) was not exposed to any stressful situation for 11 days (home cage control; HCC), while the other batch (Bl6 and 129Sv) was subjected to daily saline injections and locomotor activity measurements (repeated motility testing; RMT) for 11 days. Taking into account the differences in the action of AMPH in these two strains, the motor stimulation elicited by AMPH was studied in Bl6 and 129Sv mice in the beginning and at the end of RMT to explore the functional activity of the DA system [6]. Considering behavioral differences and vast diversity of strain-specific outcomes in pharmacological studies, we expected to see variations in transcripts of enzymes related to DA metabolism as well as DA receptors, NMDA receptor subunits, EGF family genes and their receptors in 129Sv and Bl6 strains (Figure 1, Appendix A). As we were interested in behavior implicated in the regulation of adaptation (associated with motivation and anxiety-like behavior) we decided to evaluate several brain regions. Therefore we measured gene expression from the brain structures all related with fear learning and extinction (frontal cortex, hippocampus, dorsal and ventral striatum, midbrain) [31,32,33]. Selected outcomes of gene expression in the frontal cortex and hippocampus were further validated using Western blot for protein analysis.

## 2. Materials and Methods

### 2.1. Animals

Two batches of male 129Sv and Bl6 mice were used in this study. One set of these two inbred lines (C57BL/6NTac; Taconic Germantown, New York; *n* = 25 and 129S6/SvEvTac; Taconic Germantown, New York; *n* = 26) was used as home cage controls (HCCs). The other set (C57BL/6NTac; Taconic Germantown, New York; *n* = 32 and 129S6/SvEvTac; Taconic Germantown, New York; *n* = 28) was subjected to repeated motility testing (RMT batch). Animals were bred in the local animal facility and weaned from the mother at the age of 3 weeks. Thereafter the animals were assigned to home cages with up to 10 pups per cage. Mice were housed under a 12 h light/dark cycle with lights on at 7:00 a.m. Animals were housed in their respective home cages (1290D Eurostandard type III cages; 425 mm × 276 mm × 153 mm; Tecniplast, Italy) with bedding and nesting material. The bedding (aspen chips) and nesting material (aspen wool) were changed weekly. The animals had *ad libitum* access to Ssniff universal mouse and rat maintenance diet (cat# V1534; Ssniff, Soest, Germany) and reverse osmosis-purified water, except for 1 h during testing in the RMT batch. Behavioral testing, including habituation, started at the age of 6–9 weeks, and lasted for 13 days. At the time of sample collection, animals were on average 10 weeks old.

### 2.2. Behavioral Testing

#### 2.2.1. Experimental Design for Gene Expression Studies

HCCs were weighed twice: on the 1st day and on the 11th day, right before cervical dislocation and taking brain tissue samples for gene expression measurements. The RMT batch was allocated for behavioral testing for a period of 13 days (Figure 2). The first two days were used for adaptation to the testing environment, followed by experimental days 3–13 (hereinafter days 1–11) for locomotor activity measurements. On test days 1–11 the following routine was used: animals were weighed, 0.9% saline solution was administered i.p. in volume of 10 mL/kg and animals were placed for 30 min into single housing cages (1284 L Eurostandard type II cages, 425 mm × 276 mm × 153 mm, Tecniplast, Buguggiate, Italy). After 30 min of single housing, animals were placed into the motility boxes for 30 min locomotor activity measurement and then returned to home-cages. The motor activity tests were conducted in a lit room (around 400 ± 25 lx) in soundproof photoelectric motility boxes (448 mm × 448 mm × 450 mm) made of transparent Plexiglas and connected to a computer (TSE Technical & Scientific Equipment GmbH, Berlin, Germany). After each mouse, the floor of boxes was cleaned with 5% ethanol solution. Software registered the distance travelled and the number of rearings. Latin square design was used to randomize daily measurement cycles. On day 11, immediately after the locomotor activity recordings, animals were sacrificed by cervical dislocation, decapitated and brain tissues were dissected by crude method for gene expression analysis. The anatomical coordinates of hypothalamus (mammillary bodies, optic chiasm) on the ventral surface of the brain were additionally used for orientation. Eleven-day follow-up period was chosen because after this period all the established behavioral and body weight changes had stabilized. The whole experiment can be classified as a mild stressor for mice.

#### 2.2.2. Acute AMPH Treatment Studies

The effect of acute AMPH (d-amphetamine, Sigma-Aldrich, St. Louis, MO, USA) treatment was studied in HCC and RMT. The dose of AMPH (3 mg/kg) was selected according to our preliminary studies. We used three separate groups of mice to analyze the effect of acute AMPH. For HCC: the SAL mice (acute SAL used further for repeated administration of SAL, here we used the results from the first day of repeated SAL treatment; Group 1) and acute AMPH mice (used further for repeated AMPH treatment, not analyzed in this study, here we used the results from the first day of repeated AMPH treatment; Group 2). For RMT: the RMT SAL group (the same mice as for HCC saline group; Group 1) and the RMT SAL/AMPH group (repeated treatment with SAL, followed by the administration of AMPH on the day 11; Group 3).

For HCC the first two days were allocated for adaptation to the testing environment. On the third day, AMPH was administered to mice after body weight measurement. Then the animals were placed for 30 min into single housing cages. After 30 min of single housing, the animals were placed into motility boxes for 30 min locomotor activity measurement and after calming down they were returned to their home-cages. This was necessary for avoiding fierce interaction in their home-cages injuring each other. This was apparently enough to avoid the “side-effect” of AMPH treatment. The acute AMPH mice (from day 1) were further used for repeated administration of AMPH and were not used in this study, as described above. HCC SAL (treated with 0.9% saline) mice were used further for repeated administration of SAL.

For RMT, after habituation, on the test days 1–10 mice were treated with 0.9% saline solution. On the last 11th day, mice were treated with 3 mg/kg AMPH. Then the animals were placed for 30 min in single housing and right after that into the motility boxes for 30 min locomotor measurement. After that samples were collected for further studies. Here we did not analyze the impact of AMPH on the gene expression. The locomotor activity was measured as described in 2.2.1.

### 2.3. Gene and Protein Expression Studies

For HCC, after the last body weight measurement on the 11th experimental day, and for RMT, after last locomotion activity measurement on the 11th day, mice were decapitated immediately. Decapitation and behavioral evaluations were carried out in separate rooms. The frontal cortex, hippocampus, ventral striatum, dorsal striatum, and midbrain were dissected according to the coordinates provided in the mouse brain atlas by Franklin and Paxinos and quickly frozen in liquid nitrogen [34]. All the brain tissues were stored at −80 °C. In gene expression studies, two batches of mice were measured together at the same time (both strains from HCC and both strains from RMT).

#### 2.3.1. RNA Isolation, cDNA Synthesis, and Quantitative Real-Time-PCR (qPCR)

Total RNA was extracted individually from all the brain tissues of each mouse using Trizol^®^ Reagent (Invitrogen, USA) according to the manufacturer’s protocol. RNA quality control was performed by Nanodrop where the ratios 260/230 and 260/280 were always around 2.00. After RNA extraction, DNase treatment with RNase-free DNase I (Invitrogen, Waltham, MA, USA) was applied according to the manufacturer’s protocol to prepare DNA free RNA. For the first strand cDNA synthesis, three micrograms of total RNA of each sample was used with random hexamers (Applied Biosystems, Bedford, MA, USA) and SuperScript™ III Reverse Transcriptase (Invitrogen, Waltham, MA, USA) in cDNA synthesis. In qPCR, every reaction was made in four parallel samples to minimize possible errors. All reactions were performed in a final volume of 10 μL, using 5 ng of cDNA. Real-time qPCR was performed using 5x HOT FIREPol^®^ EvaGreen^®^ qPCR Supermix (Solis BioDyne, Tartu, Estonia). For thermal cycling, a QuantStudio 12 K Flex Software v.1.2.2 Real-Time PCR System equipment (Applied Biosystems, Bedford, MA, USA) was used at 95 °C for 15 min, then 40 cycles at 95 °C for 20 s, 61 °C for 20 s and 72 °C for 20 s. All qPCR data has been presented in the log_10_ scale, in the form of 2^−ΔCT^, where ΔCT is the difference in cycle threshold (CT) between the gene of interest and housekeeper gene hypoxanthine guanine phosphoribosyl transferase (*Hprt*). For quality control, three internal samples with *Hprt* primers were used between all the measurements.

#### 2.3.2. Primer Design

Primers were designed using Primer3plus web interface [35]. Primers (LGC Biosearch Technologies, Risskov, Denmark) for qPCR assay were designed for the detection of specific transcripts from exon-exon junction, eliminating the possibility of contamination with genomic DNA (Appendix A). Melting curves were analyzed to ensure amplification specificity and no primer-dimer formation. Primer qPCR efficiency was evaluated using serial 5x dilutions of cDNA samples. *Hprt* was chosen as the housekeeper gene from previously published data and was optimized for our assay [36].

#### 2.3.3. Western Blot

Samples of the frontal cortex and hippocampus were sonicated in ice-cold RIPA lysis buffer (Thermo Scientific, Waltham, MA, USA) containing 1x protease inhibitor (78430, Thermo Scientific, Waltham, MA USA) and centrifuged at 14,000× *g* for 10 min at 4 °C. Supernatants were removed and used for Western blot analysis. Protein concentrations were measured with the BCA protein assay kit (23225, Thermo Scientific, Waltham, MA, USA) and 20 μg of protein from each sample were run on a NuPAGE Bis–Tris gel using the XCELL SureLock System (Invitrogen, Waltham, MA, USA). Protein was then transferred to a nitrocellulose membrane, after which the membranes were blocked and probed with primary antibodies (Appendix A) overnight at 4 °C. Immunoblots were then incubated with goat anti-rabbit (A11369, Invitrogen, Waltham, MA, USA) or goat anti-mouse (A-21057, Invitrogen, Waltham, MA, USA) fluorescent conjugated secondary antibodies for 1 h at room temperature, followed by visualization using a LI-COR Odyssey CLx system (LI-COR Biotechnologies, Lincoln, NE, USA). Images were converted to grayscale and the density of protein was quantified using Image Studio Lite v 3.1.4 (LI-COR Biotechnologies, Lincoln, NE, USA). B-actin was used as a loading control.

### 2.4. Statistical Analyses

Results are expressed as mean values ± SD. Statistical analyses for behavioral experiments, body weight, gene and protein expression data were performed using Statistica software (StatSoft Inc., Tulsa, OK, USA, 13th edition). Body weight change (ΔBW) was expressed as change of the initial body weight (weight on the 11th day − weight on the 1st day). For the locomotor activity in RMT, repeated measures ANOVA was applied followed by Bonferroni *post hoc* test. For amphetamine-induced locomotor stimulation one-way ANOVA was used followed by Bonferroni *post hoc* test.

To normalize the distribution of gene expression data, logarithmic transformation (log_2_) of the values was performed prior to data analysis. Comparison of gene expression data and body weight change between strains and environments was performed using two-way ANOVA [strain (Bl6 or 129Sv) × environment (HCC or RMT)] followed by Bonferroni *post hoc* test. All differences were considered statistically significant at *p* ≤ 0.05. For Western blotting, data are expressed as mean values ± SD. Differences between groups were compared with unpaired t-test with Welch’s correction or Kruskal-Wallis test. For the Kruskal-Wallis test, if a significant variance was found, the Dunn’s multiple comparison test was used for *post hoc* analysis. All figures were generated by using GraphPad software, 8th edition (GraphPad Software, San Diego, CA, USA).

## 3. Results

### 3.1. Body Weight and Locomotor Activity

In the HCC batch, the body weight was measured twice: on the 1st day and on the 11th day before collecting brain tissues. In the RMT batch, the body weight was measured on 11 consecutive days and after that the mice were exposed to the behavioral challenge in the motility box. More detailed information about the outcomes of repeated testing can be found in Appendix A.

#### 3.1.1. Body Weight Change during Experiment

In HCC batch, the body weight increased in both Bl6 and 129Sv strains, but the weight gain tended to be greater in 129Sv mice (2.19 ± 0.58 g) compared to Bl6 mice (1.39 ± 0.59 g; *p* = 0.051, Figure 3). In the RMT batch, Bl6 mice displayed almost no body weight change during 11 days (0.25 ± 0.89 g), however, 129Sv mice significantly lost weight (−1.26 ± 0.66 g; *p* ≤ 0.0001, Figure 3). The body weight differences between the strains during repeated testing appeared on the 6th experimental day (Appendix A). In both environments, 129Sv mice demonstrated greater change in body weight (2.19 ± 0.58 for HCC and −1.26 ± 0.66 for RMT, *p* ≤ 0.0001) than Bl6 (1.39 ± 0.59 for HCC and 0.25 ± 0.89 for RMT, *p* = 0.001). Subsequent application of two-way ANOVA further substantiated the differences in body weight dynamics between the two mouse strains (strain—F_(1, 42)_ = 3.04, *p* = 0.09, environment—F_(1, 42)_ = 126, *p* ≤ 0.0001, strain × environment—F_(1, 42)_ = 31.9, *p* ≤ 0.0001; Figure 3).

#### 3.1.2. Locomotor Activity in RMT Batch

The locomotor activity was measured every day. Statistical analysis was performed for the 1st and 11th experimental day. Repeated measures ANOVA for distance travelled revealed a strain effect, but no repeated testing effect (strain: F_(1,21)_ = 41.5, *p* ≤ 0.0001; repeated testing: F_(1,21)_ = 1.94, *p* = 0.32; strain x repeated testing: F_(1,21)_ = 0.16, *p* = 0.69) (Figure 4A). The distance travelled was significantly longer in Bl6 compared to 129Sv mice on the 1st day (*p* ≤ 0.0001). The motor activity of 129Sv mice tended to be higher on the 11th day, but this elevation was not significant. The difference between 129Sv and Bl6 mice on the 11th day did not differ from that established in the beginning of the study (*p* = 0.0007, Figure 4A, Appendix A). Repeated measures ANOVA for the number of rearings established a strain effect and a repeated testing effect, and their interaction was also significant (strain: F_(1,21)_ = 51.5, *p* ≤ 0.0001; repeated testing: F_(1,21)_ = 14.8, *p* = 0.0009; strain x repeated testing: F_(1,21)_ = 7.41, *p* = 0.012) (Figure 4B, Appendix A). On the 1st day, the number of rearings was significantly higher in Bl6 mice than in 129Sv mice (*p* = 0.014). The vertical activity of 129Sv mice tended to be higher on the 11th day compared to the 1st day, but this difference was not significant. By contrast, the frequency of rearings in Bl6 mice became higher with each subsequent experimental day and it robustly differed on the 11th day not only from the respective activity of 129Sv mice (*p* ≤ 0.0001), but also from their own initial activity (*p* = 0.0007; Figure 4B; Appendix A).

#### 3.1.3. Amphetamine-Induced Locomotor Stimulation

In RMT group, acute AMPH administration (3 mg/kg) in the beginning of the study (one-way ANOVA: F_(3,46)_ = 12.6, *p* ≤ 0.0001) significantly stimulated locomotor activity in Bl6 mice (*p* = 0.002 compared to saline-treated Bl6 mice; *p* = 0.007 compared to AMPH-treated 129Sv mice; Figure 5A). Also, in 129Sv mice the locomotor activity with AMPH became increased. It reached the level of saline-treated Bl6 mice, but it was not statistically significant compared to saline-treated 129Sv mice. In the end of the RMT study, the effect of AMPH (one-way ANOVA: F_(3,49)_ = 33.5, *p* ≤ 0.0001) tended to be even stronger in Bl6 strain compared to the respective saline-treated group (*p* ≤ 0.0001) and AMPH-treated 129Sv mice (*p* ≤ 0.0001; Figure 5B). In RMT 129Sv mice, the elevation of locomotor activity with AMPH again reached the level of saline-treated RMT Bl6 mice. There was a moderate difference between saline-treated Bl6 and 129Sv mice (*p* < 0.05). Also, the difference between AMPH and saline treatments in RMT 129Sv mice became statistically significant (*p* = 0.006).

### 3.2. Gene Expression Data

#### 3.2.1. NMDA and DA Systems

##### Frontal Cortex (Figure 6, Appendix A)

In HCC animals, there was a significant reduction of expression of monoamine oxidase B (*Maob*, *p* ≤ 0.0001) in 129Sv compared to the Bl6 strain (Figure 6D). RMT caused a reduction of glutamate ionotropic receptor NMDA type subunit 1(*Grin1*, *p* = 0.015), serine racemase (*Srr*, *p* ≤ 0.0001) and monoamine oxidase A (*Maoa*, *p* = 0.002) in Bl6 compared to the respective HCC group (Figure 6A–C). RMT did not change the levels of these genes in 129Sv. The strongest change with RMT in the frontal cortex was found for catechol-O-methyltrasferase (*Comt,*
Figure 6E). It was significantly upregulated in Bl6 mice (*p* ≤ 0.0001) compared to the HCC group. However, in 129Sv mice the elevation of *Comt* was even bigger and the comparison of both strains revealed a more significant increase in 129Sv strain (*p* = 0.005; Figure 6E).

##### Hippocampus (Figure 7, Appendix A)

Overall, the alterations established for 129Sv and Bl6 were more pronounced in the hippocampus compared to the frontal cortex. In HCC mice, there was a significant elevation of *Grin1* (*p* = 0.05), *Grin2b* (*p* = 0.009) and dopamine receptor D1 (*Drd1*, *p* = 0.009) in 129Sv compared to Bl6 strain (Figure 7A,B,D). Like in the frontal cortex, the expression of *Maob* (*p* = 0.005) in the hippocampus was significantly lower in HCC 129Sv than in Bl6 strain (Figure 7F). The differences between 129Sv and Bl6 established for *Grin1* (*p* = 0.027), *Grin2b* (*p* ≤ 0.0001), *Drd1* (*p* ≤ 0.0001) and *Maob* (*p* = 0.002) in HCC remained similar in the RMT group (Figure 7A,B,D,F). Besides that, RMT caused a significant elevation of *Srr* (*p* < 0.01 for Bl6 and *p* = 0.005 for 129Sv) and *Maoa* (for both *p* ≤ 0.0001) in both strains (Figure 7C,E). Only an elevation of *Maob* (*p* = 0.005) due to RMT was apparent in Bl6 strain (Figure 7F).

##### Ventral striatum (Figure 8, Appendix A)

No significant alterations of the NMDA system were found in the ventral striatum. However, again a difference between HCC Bl6 and 129Sv strains was established for *Maob* (*p* = 0.002). favoring Bl6 mice (Figure 8F). The alterations induced by RMT were similar in both strains. RMT induced a significant increase in the expression of tyrosine hydroxylase (*Th*, *p* = 0.036 for Bl6 and *p* = 0.0009 for 129Sv), *Comt* (for both *p* ≤ 0.0001), *Maoa* (for both *p* ≤ 0.0001) and *Maob* (for both *p* ≤ 0.0001) in both strains compared to the HCC group (Figure 8C–F). The expression of DA D2 receptor genes, dopamine receptor *Drd2* (*p* < 0.01 for Bl6 and *p* = 0.0007 for 129Sv) and dopamine receptor *Drd4* (*p* ≤ 0.0001 for Bl6 and *p* = 0.0003 for 129Sv) was significantly reduced in both strains in response to RMT (Figure 8A,B).

##### Dorsal Striatum (Figure 9, Appendix A)

In the dorsal striatum once again a difference between Bl6 and 129Sv in the HCC group was established for *Maob* (*p* ≤ 0.0001), favoring Bl6 mice (Figure 9F). This difference between the strains was also evident in the RMT group (*p* ≤ 0.0001). Differently from the ventral striatum, NMDA system was influenced by RMT in the dorsal striatum. RMT caused an almost similar elevation of *Grin1* (for both *p* ≤ 0.0001), glutamate ionotropic receptor NMDA type subunit 2A (*Grin2a*, for both *p* ≤ 0.0001) and *Srr* (*p* = 0.004 for Bl6 and *p* = 0.0002 for 129Sv) in both strains (Figure 9A–C). Also, the expression of DA receptors *Drd1* (*p* ≤ 0.0001 for Bl6 and *p* = 0.0003 for 129Sv) and *Drd2* (*p* ≤ 0.0001 for Bl6 and *p* = 0.0004 for 129Sv) was elevated in the same manner in both strains in response to RMT (Figure 9D,E).

#### 3.2.2. EGF Family

##### Frontal Cortex (Figure 10, Appendix A)

In HCC, the expression of only one member of the EGF family was reduced in the frontal cortex of 129Sv mice compared to Bl6 strain: transforming growth factor alpha (*Tgfa*, *p* ≤ 0.0001, Figure 10B). By contrast, RMT exposure had a larger effect on the EGF family. RMT induced an elevation of *Egf* (*p* = 0.007), *Tgfa* (*p* ≤ 0.0001), heparin binding EGF like growth factor (*Hb-Egf*, *p* = 0.0003), neuregulin 2 (*Nrg2, p* ≤ 0.0001) and erb-b2 receptor tyrosine kinase 3 (*Erbb3*, *p* = 0.005) in 129Sv compared to HCC group (Figure 10A,B,D). In Bl6 mice, RMT caused a reduction of *Nrg1* (*p* ≤ 0.0001) and a modest increase of *Nrg2* (*p* = 0.02, Figure 10C,D). The comparison of 129Sv and Bl6 in RMT group established a significant elevation of *Egf* (*p* = 0.02), *Nrg1* (*p* = 0.04), *Nrg2* (*p* ≤ 0.0001), *Erbb1* (*p* = 0.02) and *Erbb4* (*p* ≤ 0.0001) in 129Sv strain (Figure 10A,C–F). Altogether, RMT caused more pronounced alterations in the EGF family in 129Sv mice compared to Bl6 strain.

##### Hippocampus (Figure 11, Appendix A)

In the hippocampus, differently from the frontal cortex, the expression levels of *Nrg1* (*p* = 0.007) and *Erbb4* (*p* ≤ 0.0001) in HCC 129Sv mice were significantly elevated compared to the respective Bl6 group (Figure 11B,F). In RMT Bl6 the level of *Egf* (*p* = 0.0006) was reduced, whereas the expressions of *Nrg1* (*p* ≤ 0.0001) and neuregulin 3 (*Nrg3*, *p* ≤ 0.0001) were increased compared to HCC group (Figure 11A,B,D). Again, like in the frontal cortex, the changes in 129Sv mice exposed to RMT were more prominent than in the respective Bl6 group. Elevated expression levels of *Nrg1* (*p* ≤ 0.0001), *Nrg2* (*p* = 0.001) and *Nrg3* (*p* ≤ 0.0001) were evident in RMT and HCC groups in 129Sv mice (Figure 11B–D). The expressions of *Egf* (*p* = 0.01), *Nrg1* (*p* ≤ 0.0001), *Nrg2* (*p* = 0.0003), *Erbb1* (*p* = 0.01) and *Erbb4* (*p* ≤ 0.0001) were significantly higher in RMT 129Sv strain compared to RMT Bl6 (Figure 11A–C,E,F). However, one should note that differences in the levels of Nrg1 and its receptor Erbb4 were already present in HCC animals.

##### Ventral Striatum (Figure 12, Appendix A)

In the ventral striatum, the expression differences in the EGF family were less prominent compared to the frontal cortex and hippocampus. In HCC 129Sv mice, the level of *Egf* (*p* = 0.05) was reduced compared to HCC Bl6 (Figure 12A). In response to RMT, the level of *Nrg1* (*p* = 0.0002 for Bl6 and *p* = 0.003 for 129Sv) was reduced, whereas the level of *Nrg3* (*p* = 0.0003 for Bl6 and *p* ≤ 0.0001 for 129Sv) was increased in both strains (Figure 12B,C). The expression levels of *Egf* (*p* ≤ 0.0001) and its receptor *Erbb1* (*p* = 0.004) were significantly elevated if RMT and HCC 129Sv mice were compared (Figure 12A,D).

##### Dorsal Striatum (Figure 13, Appendix A)

In the dorsal striatum, the alterations of the EGF family gene expressions were rather modest like in the ventral striatum. In HCC 129Sv mice, the level of *Egf* (*p* = 0.02) was reduced compared to HCC Bl6 (Figure 13A). In RMT mice, the expression of *Nrg3* displayed a significant elevation in both strains (for both *p* ≤ 0.0001, Figure 13C). In the case of *Nrg1*, there was a modest elevation in HCC Bl6 (*p* = 0.04) compared to 129Sv, whereas in RMT this difference remained rather similar (*p* = 0.008, Figure 13B). The expression of *Egf* (*p* = 0.01) was increased in RMT 129Sv compared to respective Bl6 group (Figure 13A). In RMT 129Sv the expression levels of *Egf* (*p* ≤ 0.0001) and its receptor *Erbb1* (*p* ≤ 0.0001) were significantly increased compared to HCC 129Sv (Figure 13A,D). A similar elevation of *Erbb1* (*p* ≤ 0.0001) was established for RMT Bl6 compared to HCC group (Figure 13D).

#### 3.2.3. Gene Expression Alterations in the Midbrain (Appendix A)

Compared to the other brain regions, only a few changes occurred in the midbrain due to RMT. The expression of *Drd1* (*p* = 0.01) was reduced in RMT 129Sv compared to the HCC group (Appendix A). RMT increased the expression of dopamine receptor D3 (*Drd3*, *p* = 0.04) in 129Sv compared to Bl6. In both strains, RMT significantly decreased the level of *Comt* (*p* = 0.0007 for Bl6 and *p* = 0.01 for 129Sv).

### 3.3. Measurement of EGF Family and NMDA Protein Levels in the Frontal Cortex and Hippocampus Using Western Blot Analysis (Figure 14 and Figure 15)

In the hippocampus, the expression levels of NRG2 (*p* = 0.03) and GRIN1 (*p* = 0.002) proteins were significantly elevated in HCC 129Sv mice (Figure 14D,E).

In the frontal cortex, no statistically significant differences were established in HCC conditions. RMT exposure increased EGF family protein abundance in the frontal cortex as measured by Western blot. According to the Kruskal–Wallis test, RMT significantly increased EGF (*p* = 0.0005), ERBB1 (*p* = 0.0004), and NRG2 (*p* = 0.0014) protein levels in Bl6 compared to HCC group (Figure 15A,B,D). In addition, in the NMDA system RMT elevated protein expression of GRIN1 (*p* = 0.005) in Bl6 compared to HCC animals (Figure 15E). RMT did not affect the expression of measured proteins in 129Sv.

## 4. Discussion

Previous studies have shown that Bl6 mice display greater locomotor activity and increased exploratory behaviors while 129Sv mice are less active and are more vulnerable to stress [1,2,4]. It is worth mentioning that all 129Sv-related strains have mutated *Disc1* gene that seems to influence DA homeostasis [37,38]. Disruption of the Disc1 gene has been associated with susceptibility for various neuropsychiatric disorders, including schizophrenia [39]. Herein, we studied the impact of two different interventions (HCC and RMT) on gene and protein expression relevant for environmental adaptation. HCC represents the usual home-cage environment for the mice, whereas RMT (daily motility testing) reflects a stressful everyday challenge where mice must adapt to a stressful environment. Impaired adaptation indicates vulnerability and may thereby refer to higher susceptibility to stress-induced disorders [40,41]. Considering differences between Bl6 and 129Sv in behavior, we expected to see also variations in the gene expression of three large neurotransmitter/neuromodulator systems: DA system, NMDA receptors and EGF family. Indeed, these systems play a role in behavioral adaptation to a challenging environment in both preclinical and clinical settings [23,42,43,44,45,46,47].

### 4.1. Differences between 129Sv and Bl6 Strains in Weight Gain, Locomotor Activity and Amphetamine-Induced Hyperlocomotion

The two strains demonstrated body weight change in both conditions after 11 days. In the HCC batch, 129Sv mice gained approximately 0.8 g more body weight than Bl6. However, in the RMT batch, the strains differed in their body weight dynamics: 129Sv mice lost body weight (−1.3 g) while Bl6 mice remained almost at the initial level (+0.3 g). Taken together, 129Sv displayed a greater discrepancy in weight change when the outcomes of two interventions were compared (Figure 3). Similar results concerning differences in body weight were obtained in studies where the same strains were exposed to enriched environment [1,3].

Testing of exploratory activity also revealed differences between Bl6 and 129Sv strains. The results support previous evidence that Bl6 mice displayed remarkably higher horizontal and vertical exploratory activity compared to 129Sv mice (Figure 4, Appendix A). Initial difference in horizontal activity (the 1st day), significantly favoring Bl6 over 129Sv strain, remained the same on the 11th day. However, the difference in the frequency of rearings between Bl6 and 129Sv mice became steadily even more robust in the course of testing (Appendix A). One could suggest that Bl6 mice, differently from 129Sv strain, actively adapt to a challenging environment (trying to escape), reflected by a significantly increased number of rearings in the motility test.

Further, the differences between Bl6 and 129Sv were substantiated by treatment with AMPH. AMPH affects the function of DA transporter and increases the levels of DA in the brain [48]. In this study, treatment with AMPH caused in mice a distinct increase in locomotor activity if strains were compared (Figure 5A). In the beginning of the RMT study, the effect of AMPH in 129Sv strain reached the level of saline-treated Bl6 mice, but was not statistically different if compared to saline-treated 129Sv mice. In Bl6 mice, AMPH caused a strong elevation of distance travelled compared to vehicle treatment. These findings are in line with the study of Chen and colleagues that acute treatment with AMPH causes a significantly stronger locomotor activation in Bl6 mice compared to 129Sv strain. This behavioral effect was accompanied by augmented striatal DA efflux in Bl6 mice compared to 129Sv, whereas the basal levels of DA in these strains were not different [6]. The effect of AMPH was also evaluated at the end of the RMT study (Figure 5B). The outcome of the study was rather similar to that established in the beginning since the effect of AMPH was significantly weaker in 129Sv strain. However, here the stimulatory effect of AMPH was statistically significant in both strains if compared to respective saline treatments. Altogether, it is apparent that 129Sv and Bl6 mice display distinct sensitivity to AMPH, supporting the view about significant differences in the functional activity of DA systems in these strains.

### 4.2. Gene Expression Differences between Bl6 and 129Sv in HCC and RMT Groups

#### 4.2.1. Dopamine System

In HCC 129Sv mice, the expression of *Maob* was significantly reduced in the frontal cortex, hippocampus, ventral and dorsal striatum if compared to Bl6 strain. The values of *Maoa* and *Maob* levels in different brain structures of both strains as well as the ratio between these genes reveal that the expression of *Maoa* dominates in 129Sv mice, whereas in Bl6 strain *Maob* prevails (Appendix A; Appendix A). Enzyme MAOA preferentially oxidizes serotonin, noradrenaline, whereas MAOB preferentially cleaves phenylethylamine in mice [49]. Both enzymes are involved in the metabolism of DA. Fornai and colleagues have found that in mice DA is only metabolized by MAOA under the basal conditions and by both MAOA and B if the concentrations of DA are high [50]. Therefore, the elevated expression of *Maob* with increased response to AMPH in Bl6 mice seems to support the amplified function of DA in this strain. In line with that it has been demonstrated that Bl6 is the most sensitive strain to MPTP, the neurotoxin catalyzed by MAOB enzyme [51,52]. Interestingly, Bl6 strain is the only species where MAOB activity is greater in the brain than in liver [53].

However, the relation between Maoa and Maob was completely reversed after RMT (Appendix A). There was a tendency towards increased function of Maoa in Bl6 and Maob in 129Sv. The expression of *Comt*, another important enzyme in catecholamine metabolism, did not differ between the strains in HCC conditions.

Concerning the levels of monoamine metabolizing enzymes in RMT groups, some definite alterations occurred in the brain. The levels of *Comt* in the frontal cortex were significantly elevated in both strains in response to RMT. However, this elevation of *Comt* was more pronounced in 129Sv. In the hippocampus, the expression of *Maoa* was also increased in both strains in RMT. In the ventral striatum we found a significant elevation of all monoamine metabolizing enzymes in both strains. Besides that, the expression of *Th*, a gene for tyrosine hydroxylase, a rate limiting enzyme in DA synthesis, was elevated in both strains due to RMT. A robust elevation of all enzyme genes involved in DA metabolism (*Maoa*, *Maob*, *Comt*, *Th*) established in the ventral striatum probably underlines a pivotal role of this brain region in RMT adaptations.

*Drd1* and *Drd2* are the genes for two dominating DA receptors in the brain [54]. The only difference established in HCC mice was that in 129Sv mice the expression of *Drd1* in the hippocampus was significantly higher compared to Bl6 strain. RMT exposure even increased this difference, favoring 129Sv. The striatum comprises the dorsal striatum, which regulates motor output and decision-making, and the ventral striatum, which predominantly regulates reward and hedonic states. Both regions receive excitatory inputs from cortical and thalamic regions, as well as dense innervation from the midbrain DA-ergic nuclei [55]. DA in the ventral striatum is responsible for the exploratory drive, whereas in the dorsal striatum, the role of DA is pivotal for habit formation [56]. In the ventral striatum, a significant reduction of two DA D2 receptor family genes *Drd2* and *Drd4* was established in both strains. Taking into account that these measurements were performed immediately after the last behavioral testing and besides a decline of receptors we detected a significant increase of enzymes responsible for the metabolism of DA (*Maoa*, *Maob*, *Comt*, *Th*), one may conclude that the established changes reflect the activation of the DA system due to the behavioral challenge. One can note that the alterations in DA-related gene expression in the ventral striatum are rather similar in 129Sv and Bl6 mice. Therefore, we were not able to establish a correlation between differences in motor activity and expression of DA-related transcripts.

In the dorsal striatum, we established an elevation of *Drd1* and *Drd2* receptors in both strains due to RMT challenge. Considering that the levels of DA metabolizing enzymes were not elevated, but even reduced, like *Comt* in the dorsal striatum, this could reflect distinct alterations of DA systems in the dorsal and ventral striatum. A decrease of DA D2 receptor transcripts in the ventral striatum probably reflects ongoing exploratory drive, whereas the increase of *Drd1* and *Drd2* genes in the dorsal striatum is likely a part of habituation to the challenging environment.

#### 4.2.2. NMDA System

The comparison of NMDA receptor related gene expression in the ventral and dorsal striatum again revealed vast differences between these subcortical structures. There was no alteration in the ventral striatum, whereas in the dorsal striatum all measured NMDA receptor related genes (*Grin1*, *Grin2a*, *Grin2b*, *Srr*) were upregulated in both strains in response to RMT. This elevation was simultaneous to the upregulation of *Drd1* and *Drd2* genes in the dorsal striatum. Indeed, the dorsal striatum is rich in cortico-striatal glutamatergic projections interacting with DA [57]. The concurrent elevation of NMDA and DA receptors could be taken as a sign of intensified interaction between glutamate and DA. In the dorsal striatum GRIN1 together with DA receptors seems to play a role in the behavioral adaptation to a challenging environment [18]. In addition, Wang et al. have established a specific role of GRIN1 in habit formation [18].

In the hippocampus, the levels of *Grin1* and *Grin2b* genes were elevated in 129Sv strain compared to Bl6 mice in both HCC and RMT. In the current study we used Western blot analysis as an alternative method to measure changes in our molecular targets at the protein level. The background line specific differences in the HCC mice that were mostly detected in the hippocampal area were in line with the protein analysis: a significantly higher *Grin1* expression in 129Sv was detected both at the mRNA and protein levels, indicating elevated baseline activity of GRIN1 in 129Sv mice.

In the frontal cortex, Western blot analysis revealed an upregulation of GRIN1 protein in RMT Bl6 mice, indicating opposite dynamics between protein and transcript in stress response as levels of the *Grin1* transcript were reduced in Bl6 under in response to RMT. There are multiple factors that could explain the inconsistency in protein and mRNA levels, but miRNAs are among the most well-known regulators of transcript stability and translation efficiency. It has been shown that stress-induced miRNAs can regulate cellular responses also in the nervous system [58]. Several miRNAs control the translation efficiency in the *GRIN1* transcript making it one potential explanation of the inconsistent dynamics of GRIN1 during stress response in the two mouse lines [59].

Additional evidence for distinct functions of NMDA receptors comes from our metabolomics study where significantly increased levels of alpha-aminoadipic acid were established in the blood samples of Bl6 mice compared to 129Sv [60,61]. This difference was present in both HCC and RMT mice. Studies in rodents have shown that alpha-aminoadipic acid modulates kynurenic acid levels in the brain. Kynurenic acid is a neuroactive metabolite that interacts with GRIN1, AMPA/kainate and alpha 7 nicotinic receptors [62]. Alpha-aminoadipic acid levels dictate the availability of kynurenine aminotransferase II for the transamination of L–kynurenine to kynurenic acid [63]. Therefore, one could suggest that the formation of kynurenic acid is apparently reduced in Bl6 mice, probably causing less pronounced inhibition of GRIN1 in this strain.

#### 4.2.3. EGF Family

Research so far has established multiple interactions of EGF family proteins with DA and NMDA systems. Among the ErbB receptors, ErbB1, and ErbB4 are expressed in DA and GABA neurons, while ErbB1, 2, and/or 3 are mainly present in oligodendrocytes, astrocytes, and their precursors [44]. EGF receptor signaling upregulates the surface expression of the GRIN2B-containing NMDA receptors and contributes to long-term potentiation in the hippocampus [29]. NMDA antagonist ketamine-treated rats exhibited locomotor/stereotypy up-regulation and a defect in sensorimotor gating, resembling the behavioral phenotype of schizophrenia. Moreover, NRG1 protein levels were progressively decreased in the medial prefrontal cortex, but not in the ventral striatum of ketamine-treated rats [64].

In HCC mice, few differences in the EGF family were evident between 129Sv and Bl6 mice. *Tgfa* was significantly higher in the frontal cortex of Bl6 mice. In the hippocampus, the expression levels of *Nrg1* and its receptor *Erbb4* were significantly increased in 129Sv strain. In the ventral and dorsal striatum, the expression of *Egf* was moderately higher in Bl6 mice compared to 129Sv strain. The higher *Nrg2* in 129Sv hippocampus, previously shown as same direction trends in the corresponding transcripts, was confirmed as significant protein changes in the Western blot analysis. Altogether, the evaluation of gene and protein expression data in HCC mice shows that in 129Sv mice the activity of *Nrg1/Nrg2* dominates, whereas in Bl6 mice the activity of *Egf* tends to be higher.

In RMT mice, substantial differences between 129Sv and Bl6 strains in the EGF family were evident. As a general tendency, in the frontal cortex the expression of EGF family in RMT 129Sv mice was elevated both compared to HCC 129Sv (*Egf, Tgfa*, *Nrg2*) and RMT Bl6 (*Egf, Nrg1*, *Nrg2*, *Erbb1*, *Erbb4*) mice. One must underline a robust increase in the expression of *Nrg2* and its receptor *Erbb4* in 129Sv compared to Bl6 strain. Differently from gene expression studies, Western blot established that RMT caused a significant elevation of EGF, NRG2 and ERBB1 protein levels in the frontal cortex of Bl6 mice. This again indicates opposite dynamics between proteins and transcripts in stress response.

In the hippocampus similar tendencies in gene expression like in the frontal cortex were apparent. In RMT 129Sv mice the levels of *Egf*, *Nrg1*, *Nrg2*, *Erbb1* and *Erbb4* were increased compared to RMT Bl6 mice. However, the difference for *Nrg1* and *Erbb4* was already present in the HCC group. Comparison of RMT and HCC 129Sv mice revealed an elevated expression of *Nrg1*, *Nrg2* and *Nrg3* due to RMT. A similar tendency was evident in Bl6 mice where the levels of *Nrg1* and *Nrg3* were increased in the RMT group. The gene expression study supports the activation of *Nrg1*/*Nrg2*/*Nrg3* and *Erbb4* signaling.

In the ventral striatum, less changes occurred. The expression levels of *Egf* and its receptor *Erbb1* were increased in RMT compared to HCC 129Sv mice. On the other hand, the level of *Nrg1* was reduced and *Nrg3* was increased if HCC and RMT groups of both strains were evaluated. In the dorsal striatum, the expression of *Egf* gene was increased in RMT 129Sv mice if compared to HCC 129Sv and RMT Bl6 animals. Its receptor *Erbb1* was also elevated if RMT and HCC 129Sv mice were compared. However, the latter effect was also evident for Bl6 mice. Besides that, the expression of *Nrg1* and *Nrg3* was increased in RMT Bl6 mice compared to the HCC group. In RMT 129Sv mice we found the same effect for *Nrg3*, whereas the expression of *Nrg1* was reduced compared to RMT Bl6 mice. Therefore, in the dorsal striatum a similar upregulation, as in the case of DA receptor and NMDA related genes due to RMT, was established for several EGF family genes, including *Hb-Egf*, *Nrg3* and *Erbb1* in both strains. Altogether, data from the ventral and dorsal striatum show elevated *Egf-Erbb1* signaling in RMT 129Sv mice. Besides that, the augmented function of *Nrg3* gene is prominent in these brain structures of both strains due to RMT.

So far it is known that genetically modified mice with NRGs/ERBB receptors mutations display various behavioral alterations. Mutant mice heterozygous for either *Nrg1* or its receptor, *Erbb4*, show a behavioral phenotype that overlaps with mouse models for schizophrenia [65]. Furthermore, heterozygous *Nrg1* mice have fewer functional NMDA receptors than wild-type mice. Neonatally EGF-treated animals exhibited persistent hyperdopaminergic abnormalities in the nigrostriatal system while NRG1 treatment resulted in DA-ergic deficits in the corticolimbic DA system [22]. *Nrg2* knockouts (KO) have higher extracellular DA levels in the dorsal striatum, but lower levels in the medial prefrontal cortex (mPFC), a pattern with similarities to DA imbalance in schizophrenia [66]. Like *Erbb4* KO mice, *Nrg2* KOs performed abnormally in a battery of behavioral tasks relevant to psychiatric disorders [25]. *Nrg2* KOs exhibit novelty-induced hyperactivity in the open field, deficits in prepulse inhibition, hypersensitivity to AMPH, antisocial behaviors, and deficits in the T-maze alternation reward test—a task dependent on hippocampal and mPFC function [66]. *Nrg3* KO mice also exhibit behaviors consistent with psychotic disorders. These animals displayed novelty-induced hyperactivity, impaired prepulse inhibition of the acoustic startle response, and deficient fear conditioning [67]. Current evidence points to a central role of NRGs/ERBB receptors in controlling glutamatergic LTP/LTD (long-term potentiation/long-term depression) and GABAergic LTD at hippocampal CA3–CA1 synapses, as well as glutamatergic LTD in midbrain DA neurons, thus supporting that NRGs/ERBB signaling is essential for proper brain functions, cognitive processes, and complex behaviors [68,69].

Altogether, the most consistent gene expression findings in the EGF family were found for *Egf* and its receptor *Erbb1*, and for *Nrg1*, together with its paralogs *Nrg2* and *Nrg3*, and for their receptor *Erbb4*. In 129Sv mice RMT tended to cause an upregulation of *Nrg1/Nrg2*-*Erbb4* encoding transcripts in the frontal cortex and hippocampus. In the ventral and dorsal striatum RMT selectively increased the expression of *Nrg3* in both strains. The RMT-induced upregulation of genes encoding *Egf-Erbb1* pathway was evident in all forebrain structures in 129Sv mice.

Protein analysis in the frontal cortex confirmed RMT stress-induced differential alterations in the expression profile of EGF-related targets. However, stress-related protein profile in the two mouse lines was different from that of transcripts due to significant stress-induced upregulation of our protein targets in the Bl6 frontal cortex. In addition to the increase of NRG2 protein, also an upregulation of EGF, ERBB1 and GRIN1 proteins was detected in the frontal cortex of RMT Bl6 mice. As all line-specific protein and transcript changes were positively correlated in the hippocampus of home cage mice and inverse correlations between gene and protein expression occurred only in the frontal cortex of the RMT group, we suggest an involvement of stress-induced regulation mechanism related to the mRNA/protein stability that could be mediated through post-transcriptional processes [70]. As an example of differential post-translational mechanisms between Bl6 and 129Sv lines, it has been shown that homozygous mutation in the mRNA decay activator protein ZFP36L2 results in different phenotypes in Bl6 and 129Sv lines [71]. Additionally, in 129Sv, the inconsistency between stress-induced increase in transcripts not followed by correspondingly increased protein levels in the frontal cortex may reflect the reduced activity of EGF-family and NMDA receptor signaling in 129Sv. Increased gene expression of respective genes could be compensatory to the reduced function of these proteins in 129Sv; this assumption, however, needs further studies. Despite the directions of the stress-induced changes, the present study shows the involvement of the EGF family and its receptors in adaptation to challenging environments.

## 5. Conclusions

Bl6 and 129Sv mice displayed clearly different coping strategies in response to a challenging environment. First, we found that 129Sv demonstrated a substantially greater variation in body weight in two treatment conditions (HCC and RMT). The loss of body weight in RMT 129Sv mice is probably caused by elevated anxiety, characteristic for this mouse strain, and may reflect anxiety-induced reduction of food intake. Second, Bl6 displayed distinct active coping strategies (constantly increasing vertical activity looking for escape) in a challenging RMT environment compared with the passive approach in 129Sv. Moreover, we established a significantly stronger motor stimulating effect of AMPH, an indirect agonist on DA, in Bl6 mice. This can be taken as a sign of significantly higher functional activity of DA systems in Bl6 strain. Again, it is worth mentioning that all 129Sv-related strains have mutated *Disc1* gene influencing DA homeostasis [37,38].

DA is probably a key factor related to the differences in adaptation of 129Sv and Bl6 mice. We noticed a clear difference between the dorsal striatum and ventral striatum in response to RMT. In the ventral striatum, a substantial activation of DA metabolizing genes occurred alongside with down-regulation of DA D2 receptor genes. By contrast, in the dorsal striatum, the expression of two major DA receptors (*Drd1* and *Drd2*) was elevated without significant changes in DA metabolizing enzymes. This is probably indicative of a differential role of these brain regions in behavioral regulation. The ventral striatum is important in the control of exploratory drive, whereas habituation to the environment is likely linked to the dorsal striatum. The augmented expression of *Maob* gene, probably reflecting increased metabolism of DA, compared to *Maoa* gene in the dorsal and ventral striatum was evident in HCC Bl6 mice, but not in respective 129Sv mice [50]. Different adaptation of Bl6 and 129Sv mice to a challenging environment seems to be associated with an interplay in the frontal cortex and hippocampus. A stronger gene expression activation in response to RMT of the EGF family, especially *Egf-Erbb1* and *Nrg1/Nrg2-Erbb4* pathways, is evident in 129Sv mice compared to the Bl6 strain. Concerning the protein expression data in the frontal cortex it is likely that elevations in transcript expressions of the EGF-family is a sign of impaired function of this system in 129Sv mice. The present study revealed several ways in which differentially regulated signaling pathways related to the DA, NMDA receptor and EGF family could impact the differential behavioral adaptation in 129Sv and Bl6 mice. Altogether, considering the outcome of this study, we assume that RMT influenced Bl6 and 129Sv strains differently: for Bl6 strain RMT seemed to be a mild stressor, but for 129Sv it was rather severe. Therefore, it looks like that 129Sv is more vulnerable to stress and might be a better strain for modelling neuropsychiatric disorders than Bl6.

## Figures and Tables

**Figure 1 brainsci-11-00725-f001:**
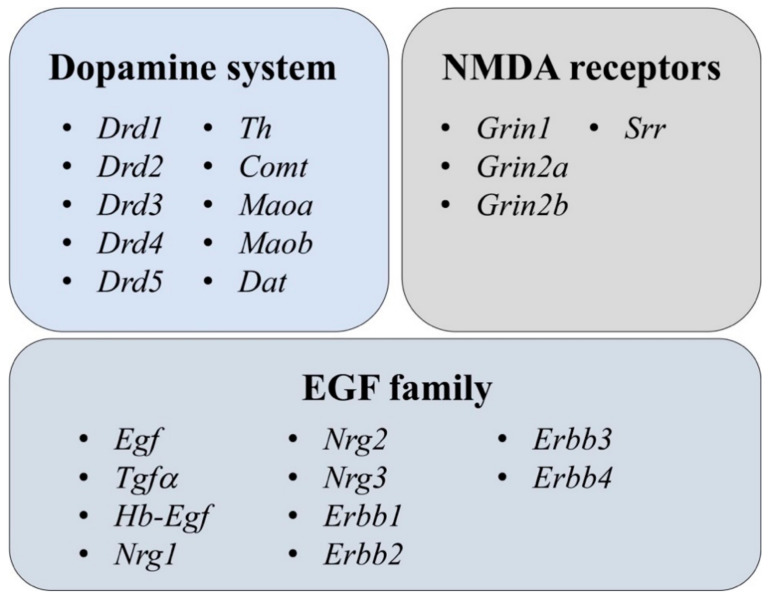
Measured gene expression from three large neurotransmitter/neuromodulator systems. More detailed information concerning full names and primers for genes is presented in Appendix A.

**Figure 2 brainsci-11-00725-f002:**
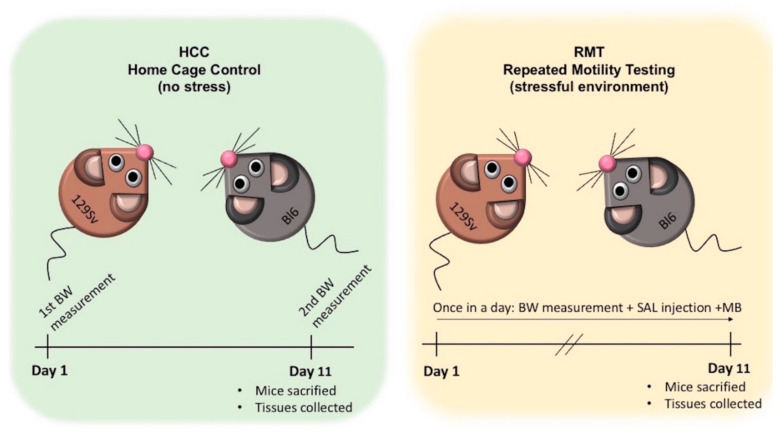
Schematic overview of the experimental design for gene expression measurement. Two batches of male 129Sv and Bl6 mice were used in this study. One batch (Bl6; *n* = 12 and 129Sv; *n* = 12) was used as home cage controls (HCCs). The other batch (Bl6; *n* = 16 and 129Sv; *n* = 14) was subjected to repeated motility testing (RMT batch). HCCs were weighed twice: on the 1st day and on the 11th day. In the RMT batch, on test days 1–11 the following routine was used: animals were weighed (BW measurement), 0.9% saline (SAL) solution was administered i.p. (SAL injection) and animals were placed for 30 min into single housing. After 30 min of single housing, animals were placed into the sound-proof motility boxes (MB) for 30 min for locomotor activity measurement and then returned to home-cages.

**Figure 3 brainsci-11-00725-f003:**
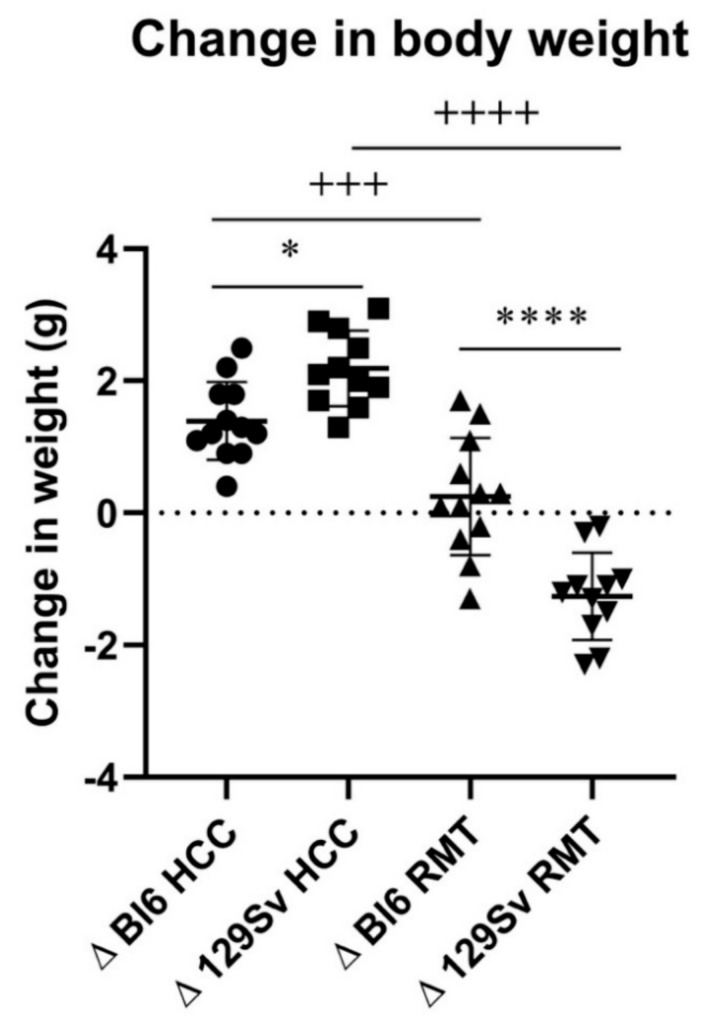
Change in body weight during the experimental period. Two-way ANOVA was applied to demonstrate differences between the strains and environments. Bonferroni *post hoc* analysis after significant two-way ANOVA: * *p* ≤ 0.05, **** *p* ≤ 0.0001 compared to respective 129Sv mice, +++ *p* ≤ 0.001 and ++++ *p* ≤ 0.0001 strain specific comparison. Data are expressed as mean values ± SD. Number of animals in each group varied from 12 to 16.

**Figure 4 brainsci-11-00725-f004:**
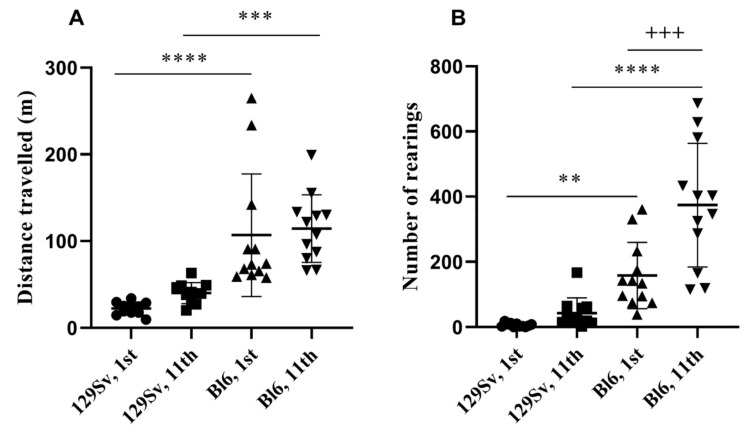
Changes in locomotor activity of Bl6 and 129Sv mice during repeated motility testing. Bl6 mice were more active both in terms of (**A**) distance travelled and (**B**) number of rearings. The difference in vertical activity between strains became augmented with each subsequent repeated testing. Data are presented as mean values ± SD. Bonferroni *post hoc* analysis after significant repeated measures ANOVA: ** *p* ≤ 0.01, *** *p* ≤ 0.001 and **** *p* ≤ 0.0001 compared to the respective 129Sv mice; +++ *p* ≤ 0.001 (compared to Bl6 on the 1st day of study). Number of animals in each group varied from 11 to 12.

**Figure 5 brainsci-11-00725-f005:**
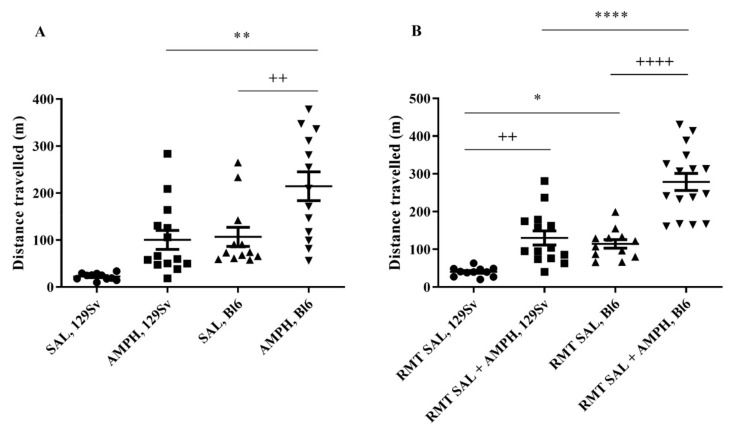
The effect of acute amphetamine (AMPH, 3 mg/kg) treatment in the beginning and at the end of RMT in Bl6 and 129Sv strains. (**A**) AMPH treatment in the beginning of study (HCC) and (**B**) AMPH treatment after repeated saline administrations (RMT). AMPH was administered 30 min before the beginning of motility test and distance travelled was measured for 30 min. Data are presented as mean values ± SD. Bonferroni *post hoc* analysis after significant two-way ANOVA: * *p* ≤ 0.05, ** *p* ≤ 0.01 and **** *p* ≤ 0.0001 compared to the respective 129Sv mice, ++ *p* ≤ 0.01 and ++++ *p* ≤ 0.0001 strain specific comparison. SAL—saline. Number of animals in each group varied from 11 to 16.

**Figure 6 brainsci-11-00725-f006:**
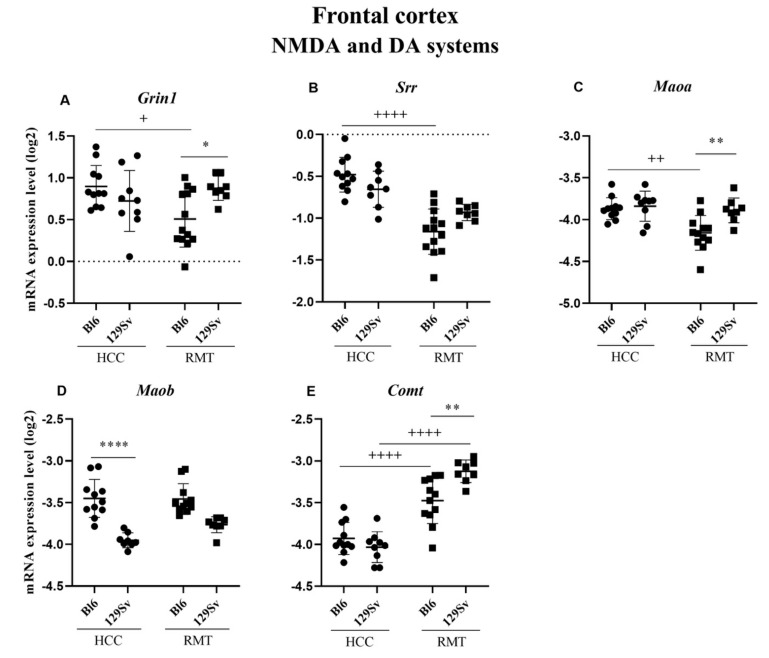
NMDA and DA systems gene expression in the frontal cortex of Bl6 and 129Sv mice. Two-way ANOVA was applied to demonstrate differences between the strains and environments. Substantial statistically significant gene expressions for (**A**) *Grin1*—glutamate ionotropic receptor NMDA type subunit 1, (**B**) *Srr*—serine racemase, (**C**) *Maoa*—monoamine oxidase A, (**D**) *Maob*—monoamine oxidase B and (**E**) *Comt*—catechol-O- methyltransferase. Bonferroni *post hoc* test: * *p* ≤ 0.05, ** *p* ≤ 0.01 and **** *p* ≤ 0.0001 compared to respective 129Sv mice; + *p* ≤ 0.05, ++ *p* ≤ 0.01 and ++++ *p* ≤ 0.0001 in strain specific comparison. Data are expressed as mean values ± SD. Number of animals in each group varied from 8 to 14.

**Figure 7 brainsci-11-00725-f007:**
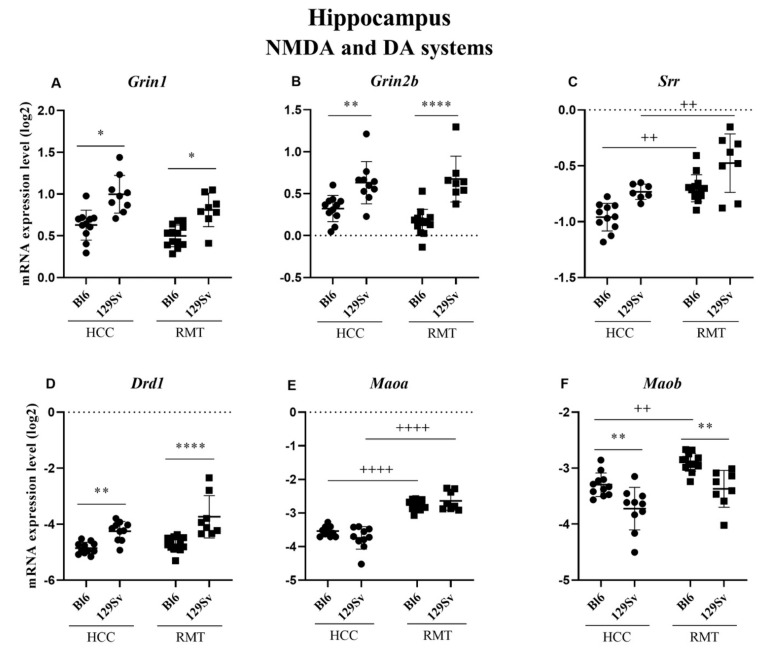
NMDA and DA systems gene expression in the hippocampus of Bl6 and 129Sv mice. Two-way ANOVA was applied to demonstrate differences between the strains and environments. Substantial statistically significant gene expressions for (**A**) *Grin1*—glutamate ionotropic receptor NMDA type subunit 1, (**B**) *Grin2b*—glutamate ionotropic receptor NMDA type subunit 2B, (**C**) *Srr*—serine racemase, (**D**) *Drd1*—dopamine receptor D1, (**E**) *Maoa*—monoamine oxidase A and (**F**) *Maob*—monoamine oxidase B. Bonferroni *post hoc* test: * *p* ≤ 0.05, ** *p* ≤ 0.01 and **** *p* ≤ 0.0001 compared to the respective 129Sv mice, ++ *p* ≤ 0.01 and ++++ *p* ≤ 0.0001 in strain specific comparison. Data are expressed as mean values ± SD. Number of animals in each group varied from 7 to 13.

**Figure 8 brainsci-11-00725-f008:**
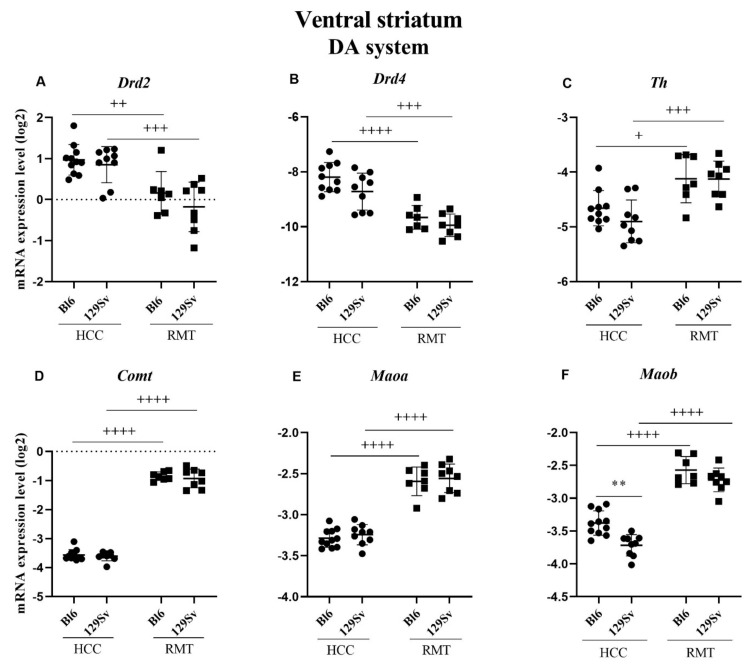
DA system gene expression in ventral striatum of Bl6 and 129Sv mice. Two-way ANOVA was applied to demonstrate differences between the strains and environments. Substantial statistically significant gene expressions for (**A**) *Drd2*—dopamine receptor D2, (**B**) *Drd4*—dopamine receptor D4, (**C**) *Th*—tyrosine hydroxylase, (**D**) *Comt*—catechol-O- methyltransferase, (**E**) *Maoa*—monoamine oxidase A and (**F**) *Maob*—monoamine oxidase B. Bonferroni *post hoc* test: ** *p* ≤ 0.01 compared to respective 129Sv mice, + *p* ≤ 0.05, ++ *p* ≤ 0.01, +++ *p* ≤ 0.001 and ++++ *p* ≤ 0.0001 in strain specific comparison. Data are expressed as mean values ± SD. Number of animals in each group varied from 7 to 11.

**Figure 9 brainsci-11-00725-f009:**
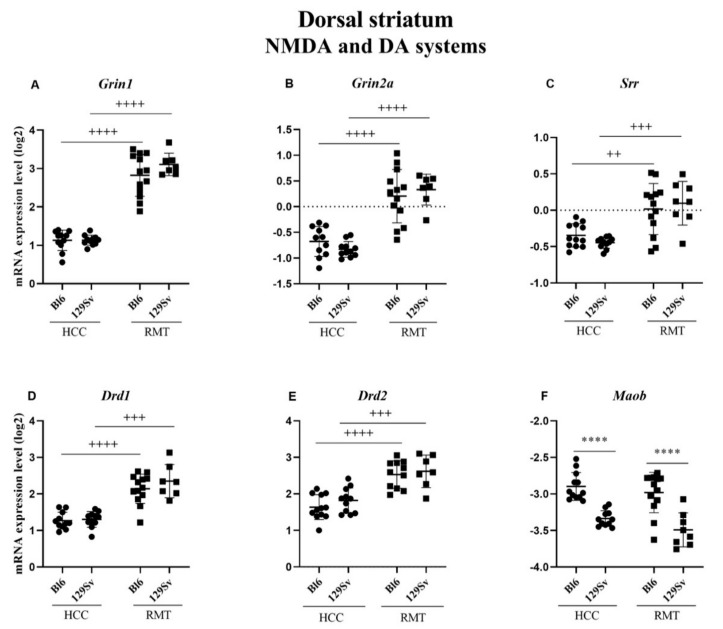
NMDA and DA systems gene expression in dorsal striatum of Bl6 and 129Sv mice. Two-way ANOVA was applied to demonstrate differences between the strains and environments. Substantial statistically significant gene expressions for (**A**) *Grin1*—glutamate ionotropic receptor NMDA type subunit 1, (**B**) *Grin2a*—glutamate ionotropic receptor NMDA type subunit 2A, (**C**) *Srr*—serine racemase, (**D**) *Drd1*—dopamine receptor D1, (**E**) *Drd2*—dopamine receptor D2 and (**F**) *Maob*—monoamine oxidase B. Bonferroni *post hoc* test: **** *p* ≤ 0.0001 compared to respective 129Sv mice, ++ *p* ≤ 0.01, +++ *p* ≤ 0.001 and ++++ *p* ≤ 0.0001 in strain specific comparison. Data are expressed as mean values ± SD. Number of animals in each group varied from 7 to 13.

**Figure 10 brainsci-11-00725-f010:**
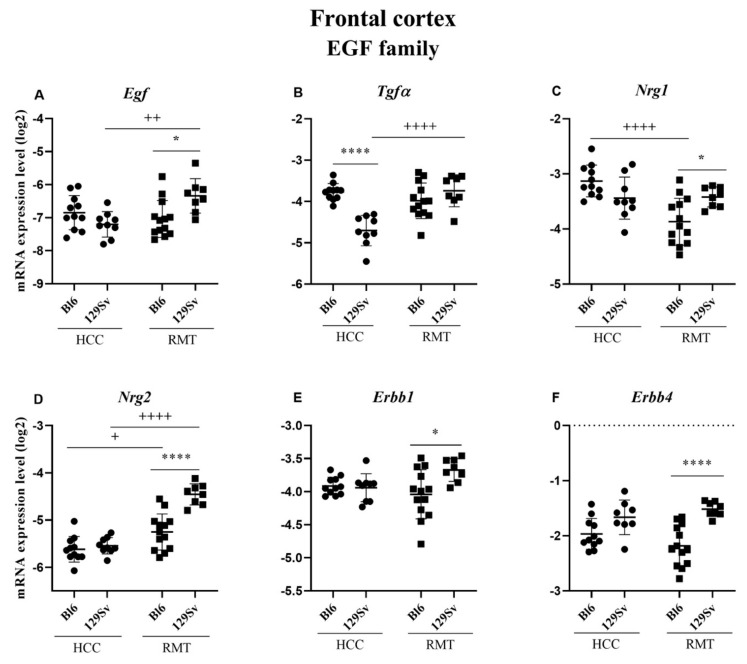
Egf family gene expression in the frontal cortex of Bl6 and 129Sv mice. Two-way ANOVA was applied to demonstrate differences between the strains and environments. Substantial statistically significant gene expressions for (**A**) *Egf*—epidermal growth factor, (**B**) *Tgfa*—transforming growth factor alpha, (**C**) *Nrg1*—neuregulin 1, (**D**) *Nrg2*—neuregulin 2, (**E**) *Erbb1*—epidermal growth factor receptor and (**F**) *Erbb4*—Erb–b2 receptor tyrosine kinase 4. Bonferroni *post hoc* test: * *p* ≤ 0.05 and **** *p* ≤ 0.0001 compared to respective 129Sv mice, + *p* ≤ 0.05, ++ *p* ≤ 0.01 and ++++ *p* ≤ 0.0001 in strain specific comparison. Data are expressed as mean values ± SD. Number of animals in each group varied from 8 to 13.

**Figure 11 brainsci-11-00725-f011:**
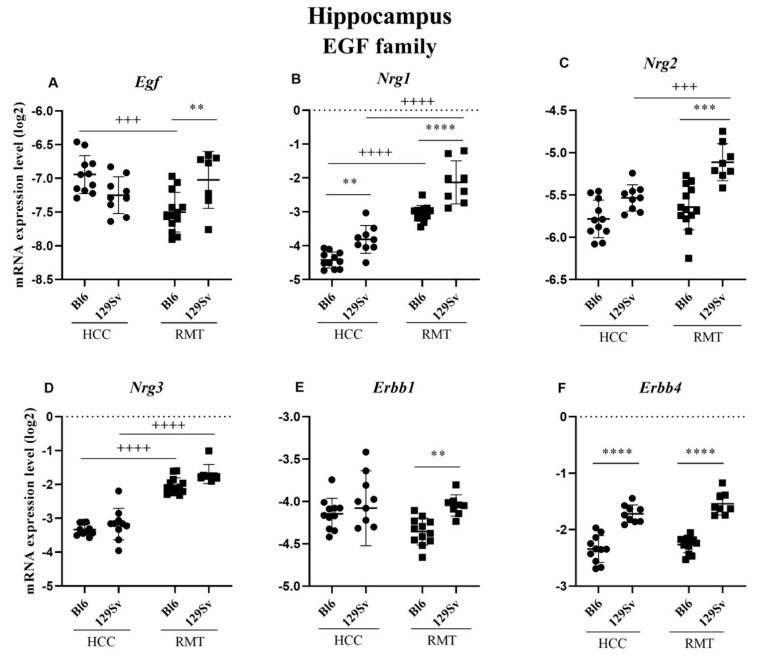
Egf family gene expression in the hippocampus of Bl6 and 129Sv mice. Two-way ANOVA was applied to demonstrate differences between the strains and environments. Substantial statistically significant gene expressions for (**A**) *Egf*—epidermal growth factor, (**B**) *Nrg1*—neuregulin 1, (**C**) *Nrg2*—neuregulin 2, (**D**) *Nrg3*—neuregulin 3, (**E**) *Erbb1*—epidermal growth factor receptor and (**F**) *Erbb4*—erb–b2 receptor tyrosine kinase 4. Bonferroni *post hoc* test: * *p* ≤ 0.05, ** *p* ≤ 0.01 *** *p* ≤ 0.001 and **** *p* ≤ 0.0001 compared to respective 129Sv mice, +++ *p* ≤ 0.001 and ++++ *p* ≤ 0.0001 in strain specific comparison. Data are expressed as mean values ± SD. Number of animals in each group varied from 7 to 13.

**Figure 12 brainsci-11-00725-f012:**
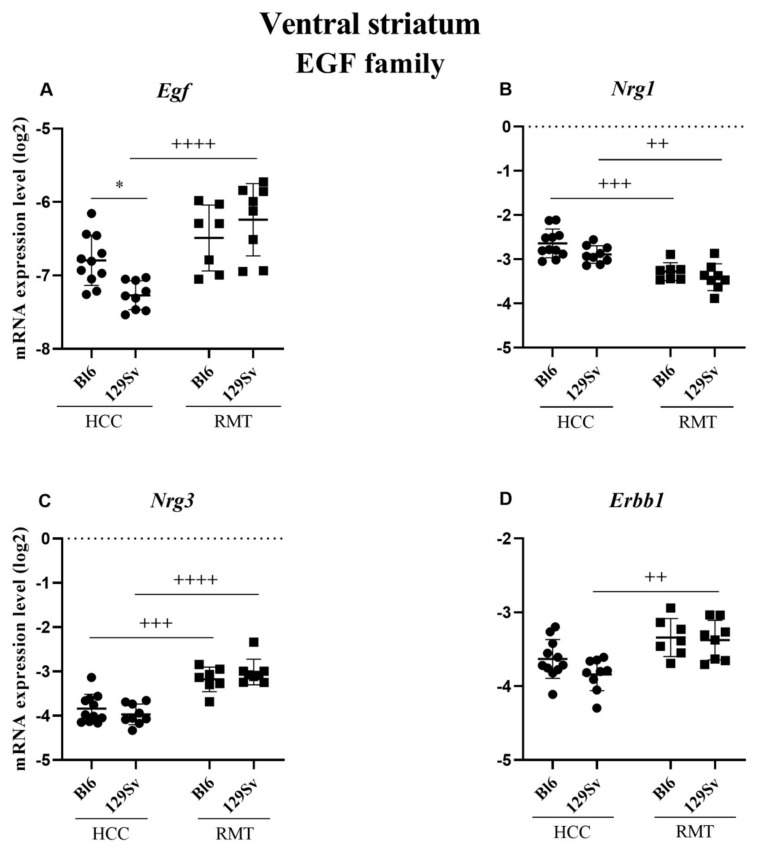
Egf family gene expression in ventral striatum of Bl6 and 129Sv mice. Two-way ANOVA was applied to demonstrate differences between the strains and environments. Substantial statistically significant gene expressions for (**A**) *Egf*—epidermal growth factor, (**B**) *Nrg1*—neuregulin 1, (**C**) *Nrg3*—neuregulin 3 and (**D**) *Erbb1*—epidermal growth factor receptor. Bonferroni *post hoc* test: * *p* ≤ 0.05 compared to respective 129Sv mice, ++ *p* ≤ 0.01, +++ *p* ≤ 0.001 and ++++ *p* ≤ 0.0001 in strain specific comparison. Data are expressed as mean values ± SD. Number of animals in each group varied from 7 to 11.

**Figure 13 brainsci-11-00725-f013:**
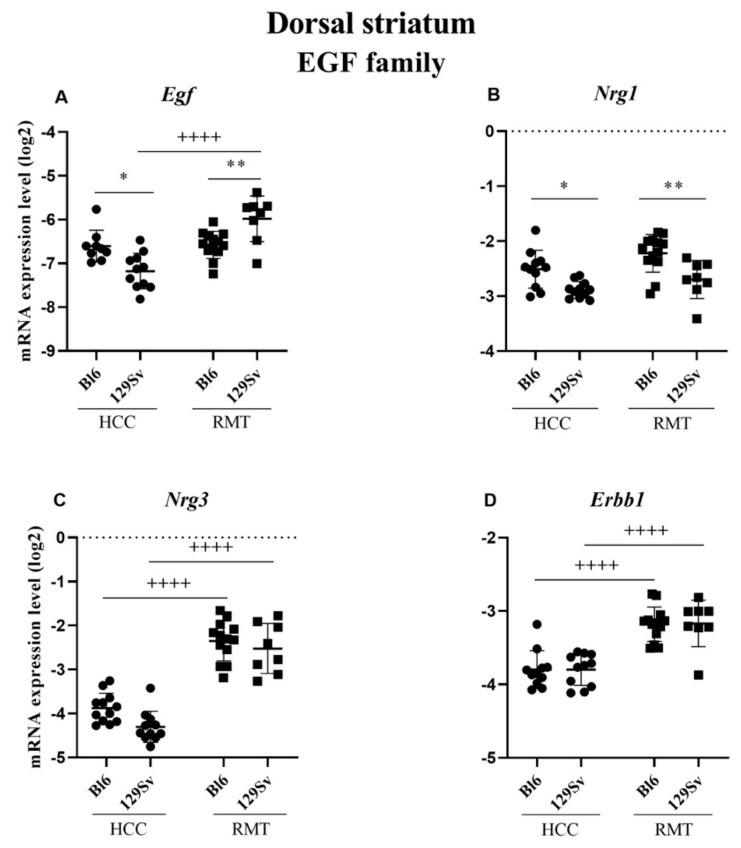
Egf family gene expression in dorsal striatum of Bl6 and 129Sv mice. Two-way ANOVA was applied to demonstrate differences between the strains and environments. Substantial statistically significant gene expressions for (**A**) *Egf*—epidermal growth factor, (**B**) *Nrg1*—neuregulin 1, (**C**) *Nrg3*—neuregulin 3 and (**D**) *Erbb1*—epidermal growth factor receptor. Bonferroni *post hoc* test: * *p* ≤ 0.05 and ** *p* ≤ 0.01 compared to respective 129Sv mice, ++++ *p* ≤ 0.0001 in strain specific comparison. Data are expressed as mean values ± SD. Number of animals in each group varied from 8 to 13.

**Figure 14 brainsci-11-00725-f014:**
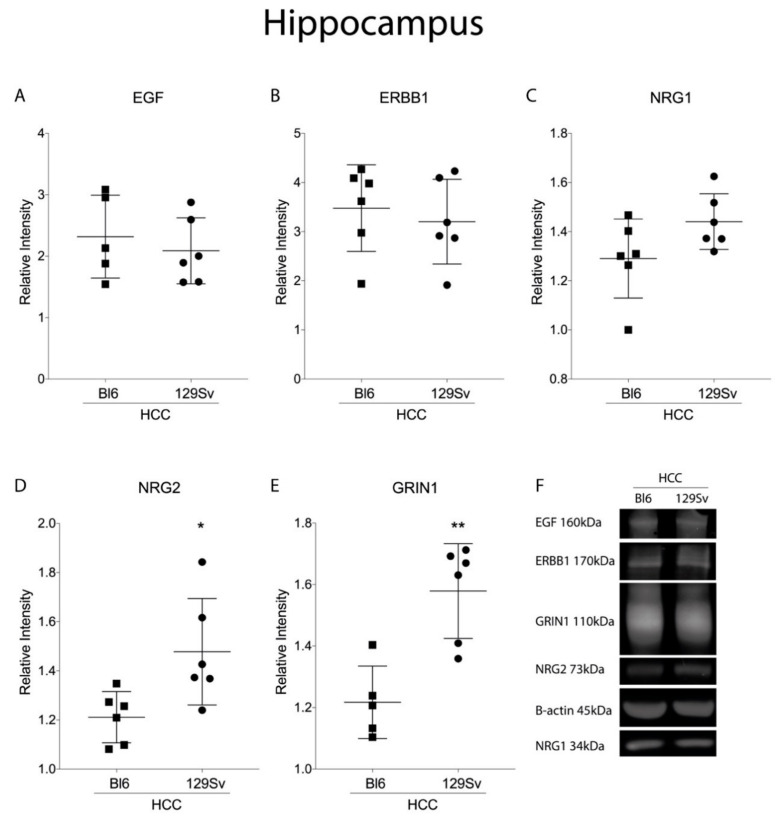
Protein expression in the hippocampus of Bl6 and 129Sv mice. Unpaired t-test with Welch’s correction was applied to demonstrate differences between the strains in HCC group. Substantial statistically significant protein expressions for (**A**) EGF—epidermal growth factor, (**B**) ERBB1—epidermal growth factor receptor, (**C**) NRG1—neuregulin 1, (**D**) NRG2—neuregulin 2, (**E**) GRIN1—glutamate ionotropic receptor NMDA type subunit 1, and (**F**) representative immunoblots. Unpaired t-test with Welch’s correction: * *p* ≤ 0.05, ** *p* ≤ 0.01 between the strains. Data are expressed as mean values ± SD. Number of animals in each group varied from 5–6.

**Figure 15 brainsci-11-00725-f015:**
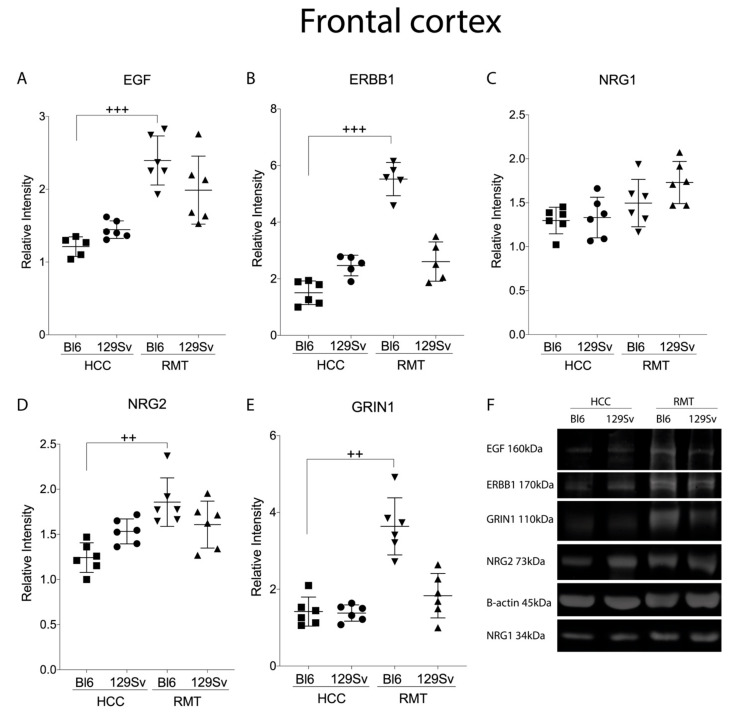
Protein expression in the frontal cortex of Bl6 and 129Sv mice. Kruskal-Wallis test was applied to demonstrate differences between the strains and environments. Substantial statistically significant protein expressions for (**A**) EGF—epidermal growth factor, (**B**) ERBB1—epidermal growth factor receptor, (**C**) NRG1—neuregulin 1, (**D**) NRG2—neuregulin 2, (**E**) GRIN1—glutamate ionotropic receptor NMDA type subunit 1, and (**F**) representative immunoblots. Dunn’s multiple comparison test: ++ *p* ≤ 0.01, +++ *p* ≤ 0.001 in strain specific comparison. Data are expressed as mean values ± SD. Number of animals in each group varied from 5–6.

## Data Availability

Not applicable.

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
