# Peer review of "Dopamine System, NMDA Receptor and EGF Family Expressions in Brain Structures of Bl6 and 129Sv Strains Displaying Different Behavioral Adaptation"

_brainsci, 2021, doi:10.3390/brainsci11060725_

Round 1

Reviewer 1 Report

In this article, the authors aim to evaluate biomarkers related to different adaptation strategies in the brain of two different strains of male mice (129Sv and Bl6 strain). Using a combination of approaches including behavioral, western blot, and RT-PCR, techniques the study reveals that BL6 and 129Sv strain mice displayed clearly different coping strategies in response to a challenging environment (RMT).

This is a very rigorous study. The overall quality is outstanding. The methodologies are very well mastered. the article is well written, the results are fairly discussed. However, I have few minor comments.

  1. Generally, I am okay with the introduction part. However, I would like the author to add rationale behind choosing the different brain areas.
  2. In the study, the author has used the western blot technique for few genes and RT-PCR for others. It would be ideal to use both the techniques for all the genes to support their findings.
  3. Social defeat stress which is another chronic stress used as a model of depression in known to increase D2 dimerization, D2 short and long-form in the prefrontal cortex in susceptible but not in unsusceptible mice (Bagalkot et al,.2015). Given that in the present study RMT seems to be a mild stressor for BL6 and a severe stressor for 129Sv. It would be interesting to see the effect of RMT on D2 receptors isoforms and dimerization between these two strains.

Author Response

1. Generally, I am okay with the introduction part. However, I would like the author to add rationale behind choosing the different brain areas.

First of all, we would like to thank you for valuable comments and suggestions. 

We have made the corresponding changes in the introduction part.

2. In the study, the author has used the western blot technique for few genes and RT-PCR for others. It would be ideal to use both the techniques for all the genes to support their findings.

Thank you for this comment. The goal of this study was to evaluate behavior related stress response and thereby we kept our focus on more rapid molecular pathways- the gene expressions. Considering the further studies we absolutely agree that additional protein measurements are necessary because we saw major strain differences between the proteins.

Currently, using western blot was limited also due to limited amount of tissue and technical reasons as not all antibodies worked properly to get straightforward quantitative outcomes.

3. Social defeat stress which is another chronic stress used as a model of depression in known to increase D2 dimerization, D2 short and long-form in the prefrontal cortex in susceptible but not in unsusceptible mice (Bagalkot et al,.2015). Given that in the present study RMT seems to be a mild stressor for BL6 and a severe stressor for 129Sv. It would be interesting to see the effect of RMT on D2 receptors isoforms and dimerization between these two strains.

This is an interesting information. A novel knowledge for us and we definitely keep this in mind.

Nonetheless, the purpose of our study was not to show the impact of serious stress. Despite that, we have tried to show that the control studies for active treatment procedures (like d-amphetamine here) induce considerable changes in behaviour and gene expression in mice. Indeed, it looked like that repeated treatment with vehicle and exposure to motility boxes caused considerable changes that were more prominent in 129Sv strain. Therefore, this kind of active/passive response/adaptation to the test conditions should be taken into account when studying repeated treatments of particular drugs in mice, especially when using different mouse strains.

Reviewer 2 Report

In this study, the author stress-tested two widely used mice strains and assessed the expression of genes in dopamine, NMDA, and EGF signaling pathways. The stress was induced by daily injection of saline coupled with repeated motility testing (RMT). Stress response in both mice strains was evaluated as a function of body weight loss over time (11 days), behavioral test evaluation, and differential regulation of signaling pathways across multiple regions in the brain. The authors note the differential response of mice strains towards the stressor environment, where 129Sv mice are more susceptible to the stress. Overall, it is good to characterize the widely used lab mice strain, highlighting the importance of background sensitivity. The result from this study can aid in the choice of mice strain for any given future research.

Major points:

1) Include background and appropriate references for RMT as a stressor model. As well as a protocol of RMT in the method section.

2) Elaborate/Discuss- Can regular RMT intervention be interpreted as exercise-associated therapeutic, which has previously shown to regulate the signaling pathways studied in this manuscript?

3) The rationale for including the acute amphetamine treatment aspect in this study is unclear.

4) Clarification required in the methods "AMPH treatment" section to whether the mice used for acute AMPH study are the same mice as those used for comparing the effect of the stressed environment across mice strains (HCC vs. RMT)? The ambiguity arises as figure legend in fig. 2 (First two sentences and last sentence of the legend), while Figure 2 does not indicate any amphetamine intervention. “Two batches of male 129Sv and Bl6 mice were used in this study. One batch (Bl6; n = 12 and 129Sv; n = 12) was used as home cage controls (HCCs). The other batch (Bl6; n = 16 and 129Sv; n = 14) was subjected to repeated motility testing (RMT batch). HCCs were weighed twice: on the 1st day and on the 11th day. In the RMT batch, on test days 1–11 the following routine was used: animals were weighed (BW measurement), 0.9% saline (SAL) solution was administered i.p. (SAL injection) and animals were placed for 30 min into single housing. After 30 min of single housing, animals were placed into the sound-proof motility boxes (MB) for 30 min for locomotor activity measurement and then returned to home-cages. In the beginning and at the end of RMT studies one group of animals received an acute injection of AMPH (3 m/kg) to evaluate the functional activity of the DA system.” It is unclear that of these 16 - Bl6 mice and 14 -129Sv mice, how many mice were used for amphetamine testing?

5) Assuming the acute amphetamine treatment conducted on a separate cohort of mice under the HCC and RMT paradigm, the day of treatment with AMPH are different for HCC (day 1 followed by SAL) and Day 11 (RMT)? Please elaborate.

6) Additionally, in the method section – acute amphetamine study – it says the study was conducted on HCC and RMT. But there is no HCC data shown in the study.

7) The methods for amphetamine treatment experiment does not match with results – i.e., Methods section – “For RMT, after habituation, on the test days, 1–10 mice were treated with 0.9% saline solution. On the last 11th day, mice were treated with 3 mg/kg AMPH.” Whereas in results - “In RMT group, acute AMPH administration (3 mg/kg) in the beginning of the study”. Please elaborate and clarify.

8) Figure 14, RMT group is missing

9) Given the key factor towards adaptation is DA regulation, protein expression data in response to stress environment in striatum should be included.

Minor Points:

1) The importance and applicability of this study can be highlighted in the abstract and elaborated in the introduction section

2) The repeated qPCR results can be represented together to minimize figures and maximize the reader’s attention. For example, Fig 6-9 can be put together by indicating the region in question next to each other, OR the data can be normalized to be represented as a heatmap with all the areas included in one heatmap.

3) All the genes indicated in Figure 1 are not included in the figures capturing various brain regions. Please indicate the reason for not including it in the result or discussion section. Additionally, some genes, such as Drd3, Drd5, Errb3, and Hb-Eg1, are not included in any of the figures. Please explain the absence.

4) Space between 2.4 and statistical analyses in the method section

Author Response

1. Include background and appropriate references for RMT as a stressor model. As well as a protocol of RMT in the method section.

First of all, we are very grateful for your time and effort to evaluate and improve our study. 

We have now hopefully added a more understandable explanation of RMT groups in the method section and have also added appropriate references for RMT. 

The peculiarity of the RMT group is that it was a control group for repeated treatment with amphetamine (Vanaveski et al., 2018). Therefore, we followed a similar routine to that used with amphetamine-treated mice. In the Vanaveski et al. paper we saw thatin response to repeated administration of saline and amphetamine, all the groups of 129Sv mice (including saline group) displayed a similar loss of body weight, an effect not seen in Bl6 mice. Accordingly, the body weight loss was not due to the repeated AMPH, but due to the stressful circumstances. Also, we have previously found that 129Sv mice have problems with coping with a novel environment in the environmental enrichment test, resulting in decreased exploratory activity and apparent weight loss (Heinla et al., 2014). 

Repeated saline administration is a very common strategy for creating a control group in  preclinical pharmacological studies. Quite often the behavioral response to the drug treatment is largely ignored in such studies. Here we identified clearly that the basic routine used in such experiments has a strong effect on behavior as well as gene expression caused by the coping with inconvenient environment. This is apparent not only in anxious 129Sv mice but also in actively adapting Bl6 strain.

Vanaveski, T.; Narvik, J.; Innos, J.; Philips, M.-A.; Ottas, A.; Plaas, M.; Haring, L.; Zilmer, M.; Vasar, E. Repeated Administration of D-Amphetamine Induces Distinct Alterations in Behavior and Metabolite Levels in 129Sv and Bl6 Mouse Strains. Front Neurosci2018, 12, 399, doi:10.3389/fnins.2018.00399.

Heinla, I.; Leidmaa, E.; Visnapuu, T.; Philips, M.-A.; Vasar, E. Enrichment and Individual Housing Reinforce the Differences in Aggressiveness and Amphetamine Response in 129S6/SvEv and C57BL/6 Strains. Behavioural Brain Research2014, 267, 66–73, doi:10.1016/j.bbr.2014.03.024.

2. Elaborate/Discuss- Can regular RMT intervention be interpreted as exercise-associated therapeutic, which has previously shown to regulate the signaling pathways studied in this manuscript?

This is a very  important question. 

We think that Bl6 mice seem to adapt with the experimental conditions better, because their behavioral activity in general is higher and elevates starting from the 3rd day of the study (Supplementary Figure S1). It seems that they are constantly looking for possibilities to escape or that they are bored with the environment, as the vertical activity increases every day. So, one may say that regular RMT intervention is an exercise-associated therapeutic for Bl6.

Regarding 129Sv strain, this phenomenon seems to be more complicated. First, the body weight decline is persistent from the day 3 to the day 11. Second, the distance travelled and number of rearings remained basically constant during the experiments. It seems that 129Sv strain reflects passive adaptation to a stressful situation and for them the RTM has no exercise-associated therapeutic effect. 

3. The rationale for including the acute amphetamine treatment aspect in this study is unclear.

It is clear that 129Sv and Bl6 mice differ by their basal activity of the dopamine system. One of the major reasons for that is the compromised function of Disc1 gene in 129Sv, playing a role in dopamine homeostasis. The only clear indication for the affected function of the dopamine system is the reduced expression of the Maob gene in all forebrain structures of HCC 129Sv mice compared to HCC Bl6. This finding may support the increased basal activity of the dopamine system in Bl6 mice. The elevated function of the dopamine system is linked to higher exploratory activity of Bl6 mice.

To support the hypothesis of the basal dopamine system activity difference (in addition to augmented expression of Maob gene in Bl6) between the strains, we treated mice with the stimulant d-amphetamine, that is an indirect agonist on dopamine. Here we established a significant difference in motor stimulating effect of d-amphetamine, strongly favoring Bl6 mice that can be taken as a sign of significantly higher functional activity of DA systems in Bl6.

 It is apparent that the elevated locomotor activity in response to amphetamine is associated with increased release of dopamine. Therefore, this is a clear marker to distinguish the activity of dopamine system in the two mice strains. 

4. Clarification required in the methods "AMPH treatment" section to whether the mice used for acute AMPH study are the same mice as those used for comparing the effect of the stressed environment across mice strains (HCC vs. RMT)? 

Thank you for bringing up this issue. All the mice used for AMPH study are not the same animals from the HCC vs. RMT stress effect study. We have made the corresponding changes to the methods section- acute AMPH treatment studies:

We used three separate groups of mice to analyze the effect of acute AMPH. For HCC: the SAL mice (acute SAL used further for repeated administration of SAL, here we used the results from the first day of repeated SAL treatment; Group1) and acute AMPH mice (used further for repeated AMPH treatment, not analyzed in this study, here we used the results from the first day of repeated AMPH treatment; Group 2). For RMT: the RMT SAL group (the same mice as for HCC saline group; Group 1) and the RMT SAL/AMPH group (repeated treatment with SAL, followed by the administration of AMPH on the day 11; Group 3).

This approach was applied to establish whether the repeated manipulations could cause a change in response to amphetamine and accordingly in the activity of dopamine system. As a matter of fact, the hyper-locomotor response to amphetamine did not markedly differ inside both strains under these two conditions (HCC and RMT), probably showing that dopamine system is not sensitized due to RMT.

The ambiguity arises as figure legend in fig. 2 (First two sentences and last sentence of the legend), while Figure 2 does not indicate any amphetamine intervention. “Two batches of male 129Sv and Bl6 mice were used in this study. One batch (Bl6; n = 12 and 129Sv; n = 12) was used as home cage controls (HCCs). The other batch (Bl6; n = 16 and 129Sv; n = 14) was subjected to repeated motility testing (RMT batch). HCCs were weighed twice: on the 1st day and on the 11th day. In the RMT batch, on test days 1–11 the following routine was used: animals were weighed (BW measurement), 0.9% saline (SAL) solution was administered i.p. (SAL injection) and animals were placed for 30 min into single housing. After 30 min of single housing, animals were placed into the sound-proof motility boxes (MB) for 30 min for locomotor activity measurement and then returned to home-cages. In the beginning and at the end of RMT studies one group of animals received an acute injection of AMPH (3 m/kg) to evaluate the functional activity of the DA system.” It is unclear that of these 16 - Bl6 mice and 14 -129Sv mice, how many mice were used for amphetamine testing?

Thank you for noticing the confusing description. The highlighted phrase should not be under the Figure 2 explanation.The sentence has now been  removed. Figure 2 explains only the experimental design for the gene expression study. As mentioned above, acute AMPH administration was the supportive experiment for describing differences in DA system activity between two strains. 

5. Assuming the acute amphetamine treatment conducted on a separate cohort of mice under the HCC and RMT paradigm, the day of treatment with AMPH are different for HCC (day 1 followed by SAL) and Day 11 (RMT)? Please elaborate.

All the mice used for acute AMPH experiment are not from the separate cohort. For example, SAL group from Figure 5A and RMT SAL group from Figure 5B are the same mice and were used also in the gene expression study. However, the AMPH mice from Figure 5A are from repeated AMPH administration group, not used in the study. RMT SAL+AMPH mice are from the acute AMPH group in RMT testing. 

To be clear, RMT SAL was a control group for repeated amphetamine administration (not used in this study), and we needed to test how such a control looks like compared to naïve mice, not exposed to repeated testing. We assumed that this RMT per se would affect the animal’s behavior and gene expression. 

6. Additionally, in the method section – acute amphetamine study – it says the study was conducted on HCC and RMT. But there is no HCC data shown in the study.

The Figure 5A results are the HCC data. We have now added this clarification to the legend of the figure. 

7. The methods for amphetamine treatment experiment does not match with results – i.e., Methods section – “For RMT, after habituation, on the test days, 1–10 mice were treated with 0.9% saline solution. On the last 11th day, mice were treated with 3 mg/kg AMPH.” Whereas in results - “In RMT group, acute AMPH administration (3 mg/kg) in the beginning of the study”. Please elaborate and clarify.

This is due to our previous not so clear interpretation regarding acute AMPH group mice definition (that has now been added to the methodology). 

“In RMT group, acute AMPH administration (3 mg/kg) in the beginning of the study”- these are the mice from a separate cohort not used in this study later (repeated amphetamine administration group and the result are taken from day 1 of this group).

8. Figure 14, RMT group is missing

Unfortunately, we do not have adequate data about the RMT group at the moment. This is due to not having enough brain material for performing of all studies. Indeed, we demonstrated signficant protein variations between the strains in HCC and, therefore, we plan to perform these studies in future, collecting enough brain material for that purpose.  

9. Given the key factor towards adaptation is DA regulation, protein expression data in response to stress environment in striatum should be included.

This is definitely an excellent idea. However, as mentioned above , using western blot was restricted due to limited amount of tissue and technical reasons, as optimisation of  some antibodies (especially dopamine transporter and dopamine D2 receptors) did not give sufficiently reliable results for quantitative estimations. Indeed, this is going a target of our further studies. 

10. The importance and applicability of this study can be highlighted in the abstract and elaborated in the introduction section

The corresponding adjustments have been made.

11. The repeated qPCR results can be represented together to minimize figures and maximize the reader’s attention. For example, Fig 6-9 can be put together by indicating the region in question next to each other, OR the data can be normalized to be represented as a heatmap with all the areas included in one heatmap.

Thank you for your valuable suggestion. However, we decided to use the initial approach because when we tried to generate the heatmap, then for our thinkig it was complicated to read out the main point of this article: the difference in gene expression between the strains in safe environment and then how the stress may influence the levels of these transcripts.

12. All the genes indicated in Figure 1 are not included in the figures capturing various brain regions. Please indicate the reason for not including it in the result or discussion section. Additionally, some genes, such as Drd3, Drd5, Errb3, and Hb-Eg1, are not included in any of the figures. Please explain the absence.

Your concern about figure representation is very accurate, however in the manuscript we added only the results with the strongest statistical outcome. Otherwise it would be very difficult to follow this in the text. All the results about gene expression with statistics are in the Supplementary Table S1. 

Additionally, when some genes had very low expression level it would be better initially exclude them from the figures. QPCR technique is a robust method and speculate something with negligible gene expression data may generate only confusion.

13. Space between 2.4 and statistical analyses in the method section

Thank you for noticing, this change has been made.

Round 2

Reviewer 2 Report

Varul et al. have addressed most of the concerns associated with the previous version of the manuscript. 

Minor suggestion: 

Below response from the authors can be added into the discussion of the manuscript. 

"Quite often the behavioral response to the drug treatment is largely ignored in such studies. Here we identified clearly that the basic routine used in such experiments has a strong effect on behavior as well as gene expression caused by the coping with inconvenient environment. This is apparent not only in anxious 129Sv mice but also in actively adapting Bl6 strain."